# HUMANLINE:
# ONLINE ALIGNMENT AS PERCEPTUAL LOSS

**Sijia Liu** [*]
Princeton University
`sijia.liu@cs.princeton.edu`

**Niklas Muennighoff** [*]
Stanford University
`muennighoff@stanford.edu`

**Kawin Ethayarajh**
University of Chicago
`kawin@uchicago.edu`

## ABSTRACT

Online alignment (e.g., GRPO) is generally more performant than offline alignment (e.g., DPO)—but why? Drawing on prospect theory from behavioral economics, we propose a human-centric explanation. We prove that online on-policy sampling better approximates the human-perceived distribution of what the model can produce, and PPO/GRPO-style clipping—originally introduced to just stabilize training—recovers a perceptual bias in how humans perceive probability. In this sense, PPO/GRPO act as perceptual losses already. Our theory further suggests that the online/offline dichotomy is itself incidental to maximizing human utility, since we can achieve the same effect by selectively training on any data in a manner that mimics human perception, rather than restricting ourselves to online on-policy data. Doing so would allow us to post-train more quickly, cheaply, and flexibly without sacrificing performance. To this end, we propose a design pattern that explicitly incorporates perceptual distortions of probability into objectives like DPO/KTO/GRPO, creating *humanline variants* of them. Surprisingly, we find that these humanline variants, even when trained with offline off-policy data, can match the performance of their online counterparts (on both verifiable and unverifiable tasks) while running up to 6x faster.

## 1 INTRODUCTION

Aligning generative models with feedback—from a human, a learned reward model, or a ground-truth verifier—is an increasingly important part of post-training, with methods categorized as offline off-policy (e.g., DPO, KTO) or online on-policy (e.g., GRPO). Despite a flurry of initial optimism around the former, recent work concurs that the latter have a higher performance ceiling, though they come at the cost of more compute, training time, and instability (Xu et al., 2024b; Ivison et al., 2024). But *why* are they better? Explanations range from online methods having better data coverage (Song et al., 2024), emphasizing generation over discrimination (Tang et al., 2024a), and navigating a simpler search space over policies (Swamy et al., 2025).

Although all these explanations have merit, we argue that if the goal is to maximize a model's utility to humans, then the dichotomy itself is incidental. We start with *prospect theory*, a framework in behavioral economics that explains why humans make decisions about random variables that do not necessarily maximize their expected value (Tversky & Kahneman, 1992). Classically, the random variable would describe a monetary outcome, measured in dollars; when extended to generative modeling, it describes the goodness of outputs, measured in bits/nats (Ethayarajh et al., 2024). Prospect theory offers a well-defined and empirically validated model of the subjective probability distribution that humans implicitly assign to outcomes. As we will show, compared to random offline data, online on-policy sampling better approximates the prospect theoretic distribution of what the model can produce, offering a human-centric explanation for why online alignment should be more performant (§3).

---

[*]Equal contribution. Work started while the last author was at Princeton Language & Intelligence.

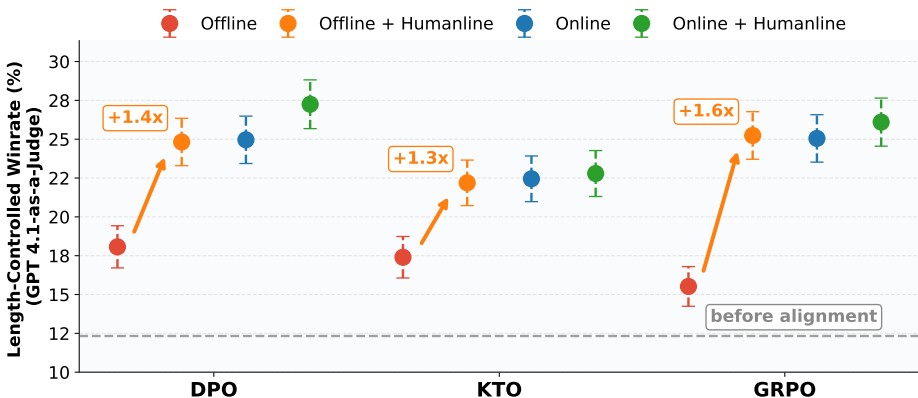

Figure 1: On instruction-following, `Llama3-8B-Instruct` aligned with online on-policy data (blue) is 1.3x to 1.6x better than one aligned with offline off-policy data (red). However, when the same offline data is fed to the *humanline* variant of the objective (orange), the gap vanishes.

However, this also suggests that online on-policy data is suboptimal on its own, as it reflects what the policy is *literally* capable of producing, as opposed to what humans *perceive* it is capable of: for example, people systematically overestimate the chance of extreme outcomes and underestimate the chance of typical ones. We then prove that PPO/GRPO-style clipping—originally introduced to just stabilize training (Schulman et al., 2017)—implicitly recovers a special case of this perceptual bias, as formalized in prospect theory. In other words, state-of-the-art alignment methods are, to some extent, perceptual losses already (§4).

If the success of PPO/GRPO can be ascribed to them being perceptual losses, then we do not necessarily need online on-policy data: we can source data from anywhere—online, offline, on-policy, off-policy—and selectively use it in a manner that reflects human perception. If we can source data from anywhere while not sacrificing performance, then state-of-the-art post-training becomes much faster and cheaper. To this end, we propose a design pattern for creating a variant of most alignment objectives (including DPO, KTO, and GRPO) that explicitly incorporates these perceptual distortions of probability while keeping the rest of the pipeline intact. This amounts to: (1) syncing the reference model with the previous version of the policy at the end of $k$ steps; (2) asymmetrically clipping the log-probability ratio of each token upstream of the loss. These simple changes, when applied correctly, create what we call the *humanline variant* of the original objective.

We consider two testbeds (§5), where an LLM is aligned to be better at: (1) instruction-following, with unverifiable rewards; (2) mathematical reasoning, with verifiable feedback. On instruction-following, an LLM trained with the online variant of DPO/KTO/GRPO has 1.3x to 1.6x higher winrates against a frontier model than one trained with the offline variant of the objective; when the same offline data is fed to the humanline variant, the gap with online alignment vanishes (Figure 1). Even on mathematical reasoning, where human utility is seemingly irrelevant, humanline GRPO allows training data to be sampled up to 64x less frequently without performance degradation.

We do not claim that you can match the performance of online alignment by simply applying the humanline variant to *any* offline data; data quality always matters. Rather, what we find is that if the data has sufficiently high average token likelihood under the reference model at the start of training, then using a humanline variant can fully bridge the gap with online alignment—this is an empirical regularity that follows from our theory. Humanline variants offer us the flexibility to source good-quality data from anywhere, which has the potential to not only make post-training many times faster and cheaper, but also make models much more adaptable to new tasks and user populations.

## 2 BACKGROUND

We provide a high-level overview of alignment methods and leave a more detailed survey to Appendix A. For the sake of brevity, we will at times refer to the online on-policy(offline off-policy) variant of a method as the online(offline) variant, in line with the literature. In most alignment al-

gorithms, including all those discussed in this paper, two copies are made of our initial model: a *reference model* $\pi_{\text{ref}}$ that serves as an anchor, whose weights are not backpropagated through; and a *policy* $\pi_\theta$ whose parameters $\theta$ are updated to minimize the loss.

**Online On-policy Alignment**    Samples are drawn from the policy, labeled with feedback—from a learned reward model, a ground-truth verifier, etc.—and fed into a loss function, which is minimized by updating $\theta$. This is done iteratively until the desired level of progress has been made, with the reference model periodically synced with the policy. The choice of loss function depends on many factors. Proximal Policy Optimization (PPO) (Schulman et al., 2017) has long been the default, since its clipped objective helps reduce training instability. Given that Grouped Relative Policy Optimization (GRPO) simplifies PPO while often improving performance (Shao et al., 2024), we use it instead. Its objective is to maximize:

$$\mathcal{L}_{\text{GRPO}}(\theta) = \mathbb{E}_{x \sim \mathcal{D}, \{y_i\}_{i=1}^G \sim \pi_{\theta_{\text{old}}}(\cdot|x)}$$

$$\frac{1}{G} \sum_{i=1}^G \frac{1}{|y_i|} \sum_{t=1}^{|y_i|} \{\min[r_\theta(i,t)\hat{A}_{i,t}, \text{clip}(r_\theta(i,t), 1-\varepsilon, \, 1+\varepsilon)\hat{A}_{i,t}] - \beta \, \text{KL}[\pi_\theta \, \| \, \pi_0]\} \tag{1}$$

where $y_i$ is an output sequence, $\theta_{\text{old}}$ is the last policy (what we call the reference[1]), $\hat{A}_{i,t} = (R_i - \text{mean}(R))/\text{std}(R)$ is the sequence-level *advantage* of output $y_i$ compared to other outputs (applied per token), $\{\epsilon, G, \beta\}$ are constants, and $r_\theta(i,t) = \pi_\theta(y_{i,t}|x, y_{i,<t})/\pi_{\theta_{\text{old}}}(y_{i,t}|x, y_{i,<t})$ is a token-wise probability ratio. KL denotes the token-wise forward KL divergence between the policy and a fixed baseline $\pi_0$ (e.g., the initial model).

**Offline Off-policy Alignment**    Online alignment is often unstable and slow, since new data needs to be continually sampled and labeled. For this reason, offline off-policy alignment has emerged as a popular alternative. Here, outputs are not drawn from the policy but from another source (e.g., human demonstrations), then fed into a closed-form loss that is minimized by updating $\theta$. The choice of loss again depends on many factors, but the most popular options are DPO (Rafailov et al., 2023), which operates on preference pairs $(x, y_w, y_l)$ where $y_w \succ y_l$, and KTO (Ethayarajh et al., 2024), which operates on unpaired feedback $(x, y_w)$ and $(x, y_l)$ (see Appendix C for precise definitions).

**Online vs. Offline**    Recent work concurs that online alignment has a higher performance ceiling, although this comes at the expense of more compute, training time, and instability (Xu et al., 2024b; Ivison et al., 2024). Recognizing their complementary strengths, some have proposed online versions of offline methods and vice-versa. For example, online DPO trains on samples generated from the latest version of the policy, closing much—but not all—of the gap with standard PPO (Guo et al., 2024; Xu et al., 2024b). Conversely, offline PPO—where the reference model is never synced and training data is static—performs similarly to offline DPO (Ethayarajh et al., 2024). Explanations of why online alignment works better have traditionally been rooted in RL theory (Song et al., 2024; Tang et al., 2024a; Swamy et al., 2025) and are thus complementary to this work.

**Verifiability**    The literature increasingly focuses on *verifiable* tasks whose correctness can be checked programmatically, such as mathematical reasoning (Lambert et al., 2025). When correctness is determined by preferences or open-ended judgments, the task is considered *unverifiable*.

## 3   ALIGNMENT AS PROSPECT THEORETIC OPTIMIZATION

Given a gamble that returns $+\$100$ with 80% probability and $-\$100$ with 20% probability, how much would a player have to be offered to forgo playing? Classical decision theory tells us that an agent meeting certain axioms of rationality (Von Neumann & Morgenstern, 1947) would have to be offered the expected value of the gamble: $0.8(+\$100) + 0.2(-\$100) = +\$60$. Most humans in this situation accept far less than $60 however, even though in expectation they could make more money gambling. *Prospect theory* offers a general framework of why, when presented with an uncertain

---

[1]Note that under our terminology $\pi_{\theta_{\text{old}}}$ would be called the reference model, since it determines the ratio $r_\theta(i,t)$, and $\pi_0$, which is called the reference in Shao et al. (2024), would be called the *baseline*. This is to ensure terminological consistency with offline methods.

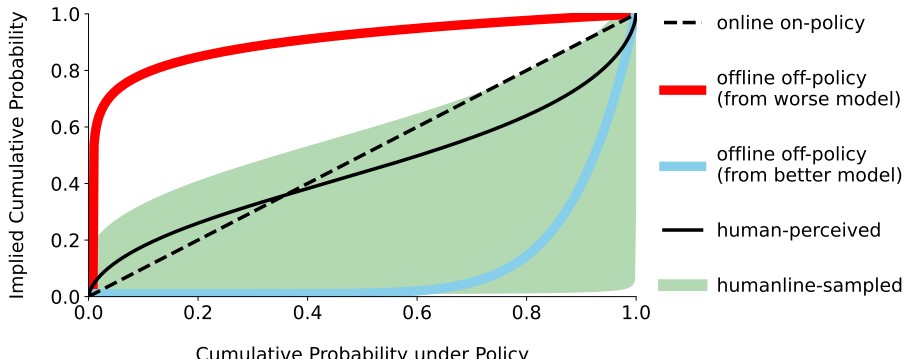

Figure 2: To estimate human utility, outputs should be sampled from the typical human-perceived distribution of what the policy can produce, whose inverted S-shape comes from *prospect theory*. Online on-policy sampling (dashed black) is superior to offline off-policy—both from worse (red) and better (blue) models—because the latter deviate more from human perception (solid black). Rejection-sampling with perceptual bias gives us *humanline sampling* (green) that can mimic this, and a special case of it simplifies to the *humanline clipping* used in our design pattern.

event, humans may choose not to maximize their expected value (Kahneman & Tversky, 1979; Tversky & Kahneman, 1992). Its model of human utility is as follows:

**Definition 3.1.** A *value function* $v : \mathcal{Z} \to \mathbb{R}$ maps an outcome $z$, relative to a reference point $z_0$, to its subjective value as perceived by the human. When $z$ is real-valued, the typical form of $v$ is:

$$v(z; \lambda, \alpha, z_0) = \begin{cases} (z - z_0)^\alpha & \text{if } z \geq z_0 \\ -\lambda(z_0 - z)^\alpha & \text{if } z < z_0 \end{cases} \tag{2}$$

where $\lambda, \alpha \in \mathbb{R}^+$ are constants.

The salient qualities of a value function are: the existence of a reference point $z_0$ used to determine the relative gain or loss; concavity in relative gains ($\alpha < 1$), known as risk aversion; and greater sensitivity to relative losses than gains ($\lambda > 1$), known as loss aversion. Under these settings, it is easy to see how the subjective expected value—as induced by $v$—could be less than \$60.

**Definition 3.2.** The *weighting function* $\omega$, when applied to an outcome $z_i$, supplants its objective probability. Let $p_i$ denote the objective probability of outcome $z_i$ and $\Omega^+$ a *capacity function* that maps cumulative probabilities to perceived cumulative probabilities. A typical functional form for the capacity function is

$$\Omega^+(a; \gamma) = \frac{a^\gamma}{(a^\gamma + (1-a)^\gamma)^{1/\gamma}} \tag{3}$$

where $\gamma \in \mathbb{R}^+$ is a constant. $\gamma = 1$ recovers the objective probability but lies in $(0, 1)$ for most humans. Letting $z_i$ denote a positive outcome relative to $z_0$ and $z_1, ..., z_n$ the ordered outcomes from least to most positive, the weights are then:

$$\omega(z_i) = \begin{cases} \Omega^+(\sum_{j=i}^n p_j) - \Omega^+(\sum_{j=i+1}^n p_j) & \text{if } i < n \\ \Omega^+(p_n) & \text{if } i = n \end{cases} \tag{4}$$

If $z_i$ were instead a negative outcome relative to $z_0$, then it would be compared to outcomes even more negative than it, with a separate function $\Omega^-(a; \gamma^-)$ following the same form as (3).

For example, suppose that in our gamble, there were now two positive outcomes instead of one: winning +\$50 with 60% probability and winning +\$100 with 20% probability. The probability of an outcome *as good or better* than \$50 is $0.8 (= 0.6 + 0.2)$ and one as good or better than \$100 is 0.2 (itself); these are our cumulative probabilities. Since $\Omega^+$ captures the human tendency to overweight extreme outcomes at the expense of moderate ones, let us say the perceived cumulative probabilities are 0.8 and 0.3 respectively. By applying (4), we then get weights of $0.5 (= 0.8 - 0.3)$ for the \$50 outcome and 0.3 for the \$100 outcome. That is, the extreme outcome of winning \$100 has been up-weighted from its objective probability of 0.2 to a subjective probability of 0.3 while the moderate outcome of winning \$50 has been down-weighted from 0.6 to 0.5.

**Definition 3.3.** The *subjective expected utility* of a random variable $Z$ is a weighted combination of the subjective values of its outcomes: $u(Z; \omega) \triangleq \sum_{z \in Z} \omega(z) v(z; \lambda, \alpha, z_0)$.

Although every human has a unique value and capacity function, the functional forms in (2) and (3) describe those belonging to the majority of people in human studies (Tversky & Kahneman, 1992).[2]

In the original literature, random variables were only studied in a monetary context, where outcomes can be measured in dollars. Ethayarajh et al. (2024) were the first to extend prospect theory to the alignment of generative models. Taking any output (token or sequence) $y$ given context $x$, they treat the surprisal term $z_{x,y} = \log[\pi_\theta(y|x)/\pi_{\text{ref}}(y|x)]$ as the outcome, whose units are nats of information. They propose that the goal of alignment is to modify $\theta$ such that desirable outputs have $z_{x,y} > z_0$ and undesirable outputs have $z_{x,y} < z_0$, formally proving that all the commonly used alignment objectives—DPO, KTO, PPO, and GRPO—encode a prospect theoretic model of utility[3], differing only in the shape of their value function and the distribution over which the expected surprisal is taken to construct $z_0$.

However, they ignore the weighting function, assuming that the human perception of probability is effectively objective when it comes to generative model outputs. But what if it were not? It is intractable to infer the human-perceived probability distribution over large output spaces (e.g., token vocabularies), which is why the original prospect theory experiments were limited to a handful of possible monetary outcomes (Tversky & Kahneman, 1992). Because of this, we will assume that the perceptual distortion of probability in the generative model setting has the same shape as in the monetary setting (Figure 2), allowing us to use the well-established parameterization in Definitions 3.1 through 3.3.

**Proposition 3.4.** *For any input $x$ and bounded value function $v$, let the outcome of an output $y$ be its surprisal $\log[\pi_\theta(y|x)/\pi_{ref}(y|x)]$ and $Q$ be a candidate distribution over outcomes. Then to guarantee $|u(Z; \omega) - u(Z; Q)| \leq \delta$ for some $\delta \geq 0$, it suffices that $\sqrt{KL(\omega\|Q)} \leq \delta/\left(\sqrt{2}\|v\|_\infty\right)$.*

The proof is deferred to Appendix B. Even if we had oracle access to a value function and thus knew exactly which alignment objective to use, we could not necessarily maximize human utility. As the proposition suggests, the simplest way to do so would be to sample generations according to the subjective distribution that was implicitly assigned to the outputs.

This offers a human-centric explanation for why online on-policy sampling is superior to offline off-policy sampling for alignment, one that is complementary to the RL-theoretic explanations in prior work. As illustrated in Figure 2, if the median human capacity function (solid black) is a function of the probabilities from the current version of the policy, then the subjective probabilities will loosely track online on-policy sampling (dashed black). In contrast, offline off-policy sampling can deviate sharply from both. Consider the desirable outputs for some context $x$:

1. When sampling from a model worse than the policy, the outputs' surprisals—computed under the current policy—will on average be lower (i.e., less positive), since they are more likely under the worse model and less likely under the current one. Recall that in (3), positive outcomes are ordered from least good to most good. This means that the implied capacity function (red) will saturate much more quickly than the human capacity function.

2. Conversely, when sampling from a model better than the current policy, surprisals will be larger. Given that more-positive outputs are more plentiful than they would be under the policy, the implied capacity function (blue) will saturate more slowly.

Returning to Proposition 3.4, if we cannot directly sample the perceived distribution, then a natural solution is to rejection-sample our outputs to simulate the drawing of tokens according to their subjective probabilities.[4] Moreover, this allows us to use data sourced from anywhere instead of limiting ourselves to online on-policy data. In §4, we modify the standard rejection sampling algorithm to capture the perceptual bias in (4), which we call *humanline sampling*. By tweaking its hyperparameters, we can mimic a wide range of distributions (Figure 2, green).

---

[2]Other parameterizations have been proposed, however: Prelec (1998); Gonzalez & Wu (1999), *inter alia*.

[3]Even losses without an explicit surprisal term, such as PPO and GRPO, do have a tokenwise likelihood ratio $[\pi_\theta(y|x)/\pi_{\text{ref}}(y|x)]$ that can be framed as the exponentiated surprisal.

[4]Although importance sampling is another option, it comes with its own problems in the context of generative models, such as degenerate importance weights.

## 4 CLIPPING RECOVERS PERCEPTUAL BIAS

In §3, we established that human utility can be maximized when outputs are drawn according to the human-perceived distribution. Given that we do not have access to anyone's perceived distribution, we will instead modify the standard rejection sampling algorithm to simulate drawing from the typical human's, as formalized by prospect theory (4). We call this *humanline sampling*.

**Proposition 4.1.** *Under typical conditions, for any context $x$, simulating output sequences $y$ from $\omega$ is equivalent to performing token-wise rejection sampling with the rejection criterion*

$$\pi_\theta(y_t|x; y_{<t})/\pi_{ref}(y_t|x; y_{<t}) < M'_\theta B$$

*where $B \sim Beta(\gamma, 1)$, $M'_\theta$ is a finite upper bound on the token-level likelihood ratio under the vocabulary (i.e., $\forall\, y_t$, $\frac{\pi_\theta(y_t|x; y_{<t})}{\pi_{ref}(y_t|x; y_{<t})} < M'_\theta$), and $\gamma \in (0, 1]$ is the capacity function constant.*

We defer the proof to Appendix B. Still, applying rejection sampling during training comes with several practical concerns. For one, in an online setting, both the reference and policy models change, and the objective probabilities that are fed into a human observer's capacity function could reflect exposure to either the current policy or the previous one. Second, resampling only those tokens that have been rejected while leaving the others untouched will not guarantee that the final output is coherent or relevant. Third, zeroing out the rejected tokens could destabilize sequence-wise losses like KTO whose training dynamics are affected by the saturation induced by all tokens. Taking this into account, we propose *humanline sampling*:

**Definition 4.2.** Given output sequence $y$, *humanline sampling* rejects tokens $y_t$ that meet the following rejection criteria by detaching them from the computational graph:

$$\frac{\pi_\theta(y_t|x; y_{<t})}{\pi_{\text{ref}}(y_t|x; y_{<t})} < M_P B_P \quad \text{or} \quad \frac{\pi_{\text{ref}}(y_t|x; y_{<t})}{\pi_\theta(y_t|x; y_{<t})} < M_R B_R \tag{5}$$

where $M_P, M_R$ are constants such that $\pi_\theta(y_t|x; y_{<t}) < M_P \pi_{\text{ref}}(y_t|x; y_{<t})$ and $\pi_{\text{ref}}(y_t|x; y_{<t}) < M_R \pi_\theta(y_t|x; y_{<t})$ for all $y_t$, $B_P \sim \text{Beta}(\gamma_P, \beta_P)$ and $B_R \sim \text{Beta}(\gamma_R, \beta_R)$ are independent Beta random variables, and $\gamma_P, \gamma_R, \beta_P, \beta_R$ are Beta distribution-specific constants.

The two-sided criteria address the first concern about the origins of the objective probabilities that are fed to a human observer's capacity functions in an online setting. Keeping the rejected tokens in the sequence addresses the second and third concerns, while detaching the tokens from the computational graph (i.e., stopping gradient flow for those tokens) ensures that they do not contribute to the updates of $\theta$ that minimize the loss. Even though we are not resampling tokens, $\gamma_R, \gamma_P$ effectively control an exploration-exploitation trade-off. If $\gamma_P < \gamma_R$, there is more emphasis on drawing from the policy (i.e., more exploitation); if $\gamma_P > \gamma_R$, there is more emphasis on exploration.

**Theorem 4.3.** *The clipped component in PPO/GRPO is a special case of humanline sampling that arises under limit conditions.*

We defer the proof to Appendix B. The intuition is that there exists a construction such that sampling from the Beta distributions is equivalent to deterministically sampling their means. The two criteria can then be combined into a range that the likelihood ratio must fall in, analogous to the clipping range. In both cases, the gradient is zero outside this range: the clipping function due to its derivative, and humanline sampling because it explicitly stops the gradients for those tokens from flowing through the graph. However, the unclipped component in PPO/GRPO does allow ratios outside this range to affect the overall gradient of the loss; to more fully integrate this perceptual bias, we would need to clip the ratios upstream of the objective, not just within it (§5).

## 5 HUMANLINE VARIANTS

### 5.1 METHOD

If the success of PPO/GRPO can be ascribed to them being perceptual losses (§3, §4), then we need not limit ourselves to using online on-policy data; we can source data from anywhere and selectively train on it in a manner that reflects the prospect theoretic model of perceived probability. To this

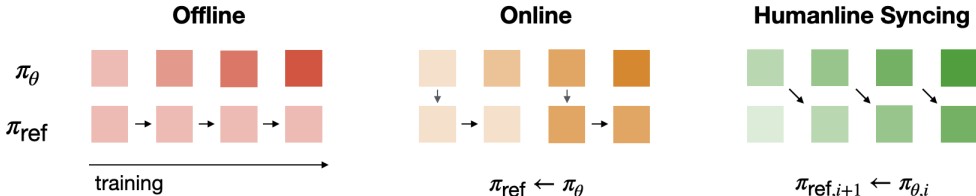

Figure 3: In offline objectives (left), the reference model does not change during training. In online objectives (middle), the reference is synced with the policy at the *current* step; at scale, some asynchrony is permitted (a lag of one step is depicted here). In *humanline syncing* (right), every $k$ steps, the reference is synced with the policy from the *previous* step ($k = 1$ is depicted here).

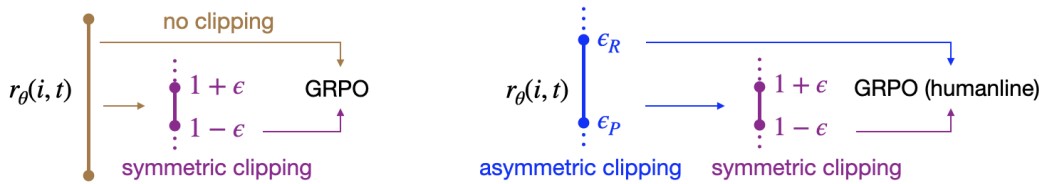

Figure 4: In *humanline clipping*, the token-wise likelihood ratios $r_\theta(i, t)$ are asymmetrically clipped to $[\epsilon_P, \epsilon_R]$ upstream of the loss. In the humanline variant of GRPO, instead of there being an unclipped $r_\theta$ and a $[1 - \epsilon, 1 + \epsilon]$-clipped $r_\theta$ as in (1), we have a once-clipped and twice-clipped $r_\theta$. Though humanline clipping should in theory be most impactful for losses without any clipping to begin with (e.g., DPO, KTO), it still benefits GRPO (see Figure 5, left).

end, we propose creating a *humanline variant* of any alignment objective that is a function of both a policy $\pi_\theta$ and reference model $\pi_{\text{ref}}$.[5] This is done by applying a two-part design pattern:

1. **Humanline Syncing**: Every $k$ steps, after the loss is calculated but before the optimizer step is taken, sync the weights of $\pi_{\text{ref}}$ with $\pi_\theta$ (Figure 3). In general, lower $k$ leads to better performance but also more instability (Figure 8, Appendix D).

2. **Humanline Clipping**: Clip all token-wise likelihood ratios $\pi_\theta(y_t|x, y_{<t})/\pi_{\text{ref}}(y_t|x, y_{<t})$ to the range $[\epsilon_P, \epsilon_R]$ even *before* they are fed into the loss, where $\epsilon_P, \epsilon_R \in \mathbb{R}^+$ and the range can be asymmetric. Losses that already do some clipping, such as GRPO, will do clipping twice over (Figure 4). We clip in log-space for greater numerical precision.

The motivation behind humanline syncing is that as the policy changes over the course of training, the standard against which the policy is judged also changes. Since the outcome is defined as the surprisal $\log[\pi_\theta(y|x)/\pi_{\text{ref}}(y|x)]$, this means that the reference model must change as well, at a rate controlled by $k$. We choose to implement humanline clipping instead of the humanline sampling proposed in §3 for a few different reasons. For one, humanline clipping is a special case of humanline sampling, one that arises under limit conditions (Theorem 4.3). However, it is much faster (since no new tensors have to be allocated), requires fewer hyperparameters, and is more stable than humanline sampling while being as or more performant (Figure 5, right). Clipping multiple times (Team et al., 2025) and asymmetric clipping (Yu et al., 2025) have been explored in past work, but to our knowledge, the specific formulation in humanline clipping has not been used.

Note that the humanline variant of each method can be used with both online on-policy data and offline off-policy data, which we denote as *online+humanline* and *offline+humanline* respectively. In contrast, the online variant of a method is only used with online on-policy data; the offline variant, only with offline off-policy data. Alignment objectives without a reference model, such as SimPO (Meng et al., 2024), cannot have a humanline variant because neither change is applicable.

---

[5]As defined in §2, the reference model is the one against which the surprisal is calculated (explicitly, in the log-ratios of DPO and KTO; implicitly in the ratios of PPO and GRPO).

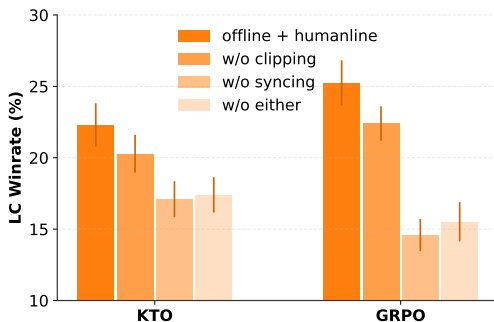 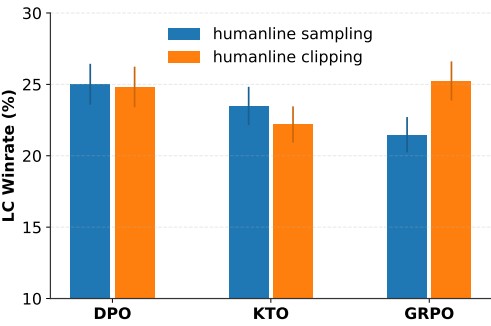

Figure 5: The majority of the improvement comes from *humanline syncing* (left). However, *humanline clipping* is still necessary—syncing alone is not competitive with online alignment. Although humanline clipping is a special case of the more general *humanline sampling* (§4), it performs as well while being stabler and simpler to implement (right).

## 5.2 EXPERIMENTS

We create humanline variants of DPO/KTO/GRPO and compare them to both their offline and online counterparts, ensuring that the number of examples seen by the different variants is the same. Details on how we created the online version of DPO/KTO and the offline version of GRPO can be found in Appendix C. We test these variants in an *unverifiable* reward setting where the goal is to follow open-ended instructions and a *verifiable* reward setting where the goal is to do mathematical reasoning.

### 5.2.1 UNVERIFIABLE REWARDS

Using offline DPO/KTO/GRPO, we first align `Llama3-8B-Instruct` (AI@Meta, 2024) on an instruction-following dataset called UltraFeedback ArmoRM (Meng et al., 2024). For online DPO/KTO/GRPO, we use the same contexts but sample completions from the policy, score them with the ArmoRM reward model (Wang et al., 2024), and then construct preference pairs. This is how the offline data was constructed as well—but sampled from different models—allowing for an apples-to-apples comparison between the online and offline variants of the same objective. Using the humanline variants with either the online or offline data does not require further changes. The models are evaluated with AlpacaEval2 (Dubois et al., 2024).[6]

**For all objectives, the offline+humanline variant performs significantly better than the offline variant** ($p < 0.05$)[7] **and is on par with the online variant**, as seen in Figure 1. The magnitude of improvement is large, with offline+humanline GRPO performing 1.6x better than its offline counterpart. Improvements persist at the 27B scale and with different model families (Appendix D). However, the online+humanline variants are only slightly better than their online counterparts. This is not surprising: under our theory (§3), online on-policy sampling is superior to offline off-policy sampling because it deviates far less from human perception; the marginal benefit of humanline objectives will naturally be smaller in this case.

**Humanline objectives do not obviate the need for good-quality data.** We stress that although offline+humanline variants *can* match the performance of their online counterparts, this is not a given for *any* offline data (Appendix D, Table 5). Fortunately, we find that the average token log-probability of the output under $\pi_{\text{ref}}$ (before training starts) is a good proxy for whether the offline data will be 'good enough'. Training on the lowest quartile—with average token log-probability in the range $[-1.03, -0.36]$—leads to significantly worse results than training on the rest (Figure 6). This can be ascribed to lower sample efficiency that arises from more frequent humanline clipping.

**Humanline syncing is responsible for most of the improvement; humanline clipping is needed to fully close the gap.** In Figure 5, we plot the drop in performance as one or both changes are ablated. Humanline syncing, done here every step ($k = 1$) is the more crucial ingredient; without it, the performance would be as bad as with the offline variant. However, it can be done as infrequently

---

[6]We use GPT 4.1 as the judge instead of the default GPT-4-Turbo, as it is cheaper and more performant.

[7]We apply the Holm-Bonferroni correction to adjust for multiple comparisons (Holm, 1979; Dunn, 1961).

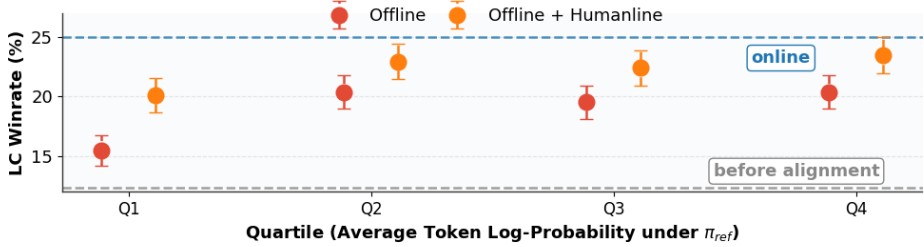

Figure 6: Data quality matters, even when using humanline variants. As seen here, the average token log-probability of the output under $\pi_{\text{ref}}$ (at step 0) is a good proxy for offline data quality; if it is too low, as in the first quartile of data, the DPO-aligned model's performance will be worse.

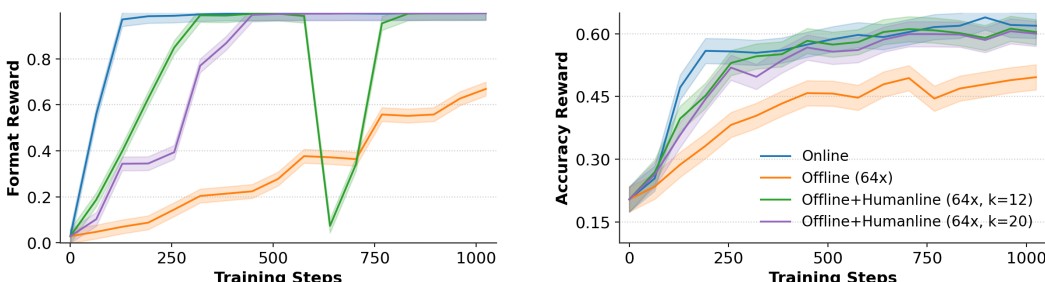

Figure 7: For mathematical reasoning (MATH500), sampling data 64x less frequently (orange) than in online GRPO (blue) leads to significantly worse performance, even though the total volume of data seen remains the same. In contrast, using the humanline variant of GRPO while being 64x more offline does not incur performance degradation (green). Less frequent humanline syncing ($k = 20$, violet) leads to slower but more stable learning; at $k = 1$, the instability would cause collapse.

as $k = 4$ without a loss in performance (Appendix D, Figure 8). Note that not all kinds of syncing are equal: trust region-style syncing (Gorbatovski et al., 2024), which happens *after* the policy is updated—thus rendering the policy and reference equal—leads to worse results (Appendix D, Figure 10). Humanline clipping is still needed for the offline+humanline variants to match the performance of their online counterparts (Figure 5, left). For instruction-following, a clipping range of $\log \epsilon_P = -1.5, \log \epsilon_R = 1.5 \iff \epsilon_P = 0.22, \epsilon_R = 4.48$ works best for the humanline variants of all methods, and performance is robust to small changes (Appendix D, Table 6).

**Humanline variants do not require changing method-specific hyperparameters, but the learning rate or maximum gradient norm need to be adjusted.** The use of a humanline variant introduces two counteracting forces. On one hand, the likelihood ratios $r_\theta$ can get smaller compared to offline learning—explicitly, due to clipping, and implicitly, due to the syncing of the reference model, since $\pi_\theta(y|x)$ cannot drift too far from $\pi_{\text{ref}}(y|x)$—causing gradients to get smaller. The learning rate or maximum gradient norm needs to increase to make up for this. Conversely, updating the reference model introduces more training instability, which demands a lower learning rate or maximum gradient norm. Therefore, depending on the circumstances, this shift could require increasing or decreasing the learning rate/gradient norm by 0.1x–4x (Appendix D, Table 3).

**Offline+humanline GRPO is over 6x faster to train with than the online variant, while attaining equal performance.** Compared to offline GRPO, the offline+humanline variant takes roughly twice as long to run when syncing every step (Appendix D, Figure 11). However, this is a comparatively small price to pay to match the performance of online GRPO, which takes over 12x the wall-clock time of offline GRPO.

### 5.2.2 VERIFIABLE REWARDS

When doing alignment for mathematical reasoning, it is standard to be fully online on-policy and use the correctness of the final output as the only reward (DeepSeek-AI et al., 2025). Our goal

with the humanline variants will be to push the extent to which the data can be offline off-policy. For example, sampling completions every 10 steps instead of every step would make the process much more efficient: in the fully online on-policy setup, training waits on the next batch of samples from the current policy and inference requires the policy to finish training on the current batch; by sampling less frequently, training, inference, and labeling can all be asynchronously overlapped.

**Humanline GRPO allows data to be sampled up to 64x less often with no performance degradation on mathematical reasoning.** We first align `Qwen2.5-1.5B-Instruct`[8] (Yang et al., 2025) with online GRPO on the MATH500 training set (Lightman et al., 2023), largely following the setup in `Open-R1` (Hugging Face, 2025) and assigning rewards based on formatting and correctness. Instead of sampling every step, we then sample 64 times as much data every 64 steps to get a model that is significantly worse ($p < 0.05$) (Figure 7). Running the same off-policy setup with the humanline variant of GRPO closes the gap in rewards within 1000 steps. After 1600 steps, the Pass@1 accuracy on the MATH500 test set is $0.593 \pm 0.019$ for both the online and humanline runs. The degree of humanline clipping remains the same as in instruction-following ($\log \epsilon_P = -1.5, \log \epsilon_R = 1.5$), suggesting that it works as a strong default for a wide variety of tasks. However, we find that syncing too frequently ($k = 1$) leads to reward collapse. Increasing $k$ leads to slower but stabler training, with any $k \in [12, 24]$ closing the gap with online alignment in 1000 steps while avoiding collapse. Although human utility seems irrelevant to mathematical correctness, the fact that reasoning is still expressed in language, a human abstraction, may help explain why incorporating perceptual biases via a humanline objective is still useful for this task.

## 6 LIMITATIONS & FUTURE WORK

We stress that although humanline variants trained with offline off-policy data are able to match the performance of their online counterparts, this is still an empirical regularity as opposed to a formal guarantee. In addition to the average token log-probability of the output under $\pi_{\text{ref}}$, are there other metrics that we can use to quantify what makes offline data 'good-quality'? Conversely, are there settings under which alignment data must necessarily be online and on-policy? We leave these as directions for future work.

The model of human utility discussed in this paper comes directly from prospect theory, which was originally developed in the context of monetary random variables. Although it has since been empirically validated in other contexts, there is no guarantee that it naturally extends to the generative modeling setting. Assuming that it does is another limitation of our work, one we accept because our primary goal is to improve the post-training of generative models, and because experimentally inferring biases in human perception over very large output spaces is intractable. Developing new theories of human probability perception as it relates to generative models is another future direction.

Lastly, humanline variants raise practical questions: How large are the systems gains from fully overlapping training/inference/labeling? Can we reduce the cost of syncing (e.g., by only syncing some of the model weights)? Should $\gamma$ be personalized instead of using one setting for all?

## 7 CONCLUSION

Based on a prospect theoretic framework, we proposed that the online-offline dichotomy central to post-training is incidental to actually maximizing utility: what matters is not the source of data *per se*, but whether it reflects the human-perceived distribution over model outcomes. This perspective interprets PPO/GRPO's clipping as recovering a form of probability distortion, suggesting that these state-of-the-art objectives are successful because they are perceptual losses. We then proposed a generic design pattern for explicitly incorporating perceptual biases into commonly used alignment objectives, giving us *humanline variants* of DPO/KTO/GRPO. When trained with offline off-policy data, the humanline variants were able to match the performance of their online counterparts, closing 1.3–1.6x gaps in winrate for instruction-following and enabling up to 64× less frequent sampling in mathematical reasoning without performance degradation. This opens the door to cheaper, faster, and more parallelizable alignment that is not constrained by the need for online on-policy data.

---

[8]Since mathematical reasoning on MATH500 requires the generation of many intermediate reasoning tokens, we were forced to use a smaller model than in §5.2.1 due to memory constraints.

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

# A RELATED WORK

**Alignment Methods**   Reinforcement Learning from Human Feedback (RLHF) involves training a reward model on human preference data and then using it to fine-tune a policy, commonly via online reinforcement learning (Christiano et al., 2017; Schulman et al., 2017; Nakano et al., 2021; Ouyang et al., 2022). The complexity of online RL has motivated a line of research on simpler, offline methods that optimize a policy on a static dataset (Ziegler et al., 2019; Rafailov et al., 2023; Ethayarajh et al., 2024; Hejna et al., 2023; Azar et al., 2024; Hong et al., 2024; Munos et al., 2023; Xu et al., 2024a; Jung et al., 2024; Wu et al., 2024a; Xie et al., 2024; Pang et al., 2024; Tang et al., 2024b). Other work aims to bridge the gap between offline and online methods via iteratively collecting new data from the policy (Stiennon et al., 2020; Xu et al., 2023; Xiong et al., 2023; Wu et al., 2024b; Chen et al., 2024; Rosset et al., 2024; Pang et al., 2024; Lanchantin et al., 2025b), reweighting offline loss terms (Zhou et al., 2024), or recasting offline methods as online (Guo et al., 2024). Recently, there has been an increase in interest in online reinforcement learning with verifiable rewards, including training hyperparameters (Yu et al., 2025; Liu et al., 2025; Wang et al., 2025a) and other aspects (Wang et al., 2025b; Zuo et al., 2025; Shao et al., 2025; Yue et al., 2025; Zhao et al., 2025). The humanline design pattern can be applied to most alignment algorithms in both offline and online settings.

**Sources of Feedback**   The performance of alignment algorithms is directly linked to the type and quality of the feedback signal. This signal often comes from direct human judgment (Bai et al., 2022a; Wu et al., 2023). To improve scalability, researchers have also explored the use of AI-generated feedback (RLAIF) (Bai et al., 2022b). For more objective domains such as coding, verifiable feedback can be derived from execution results and unit tests (Le et al., 2022; Liu et al., 2023; Gehring et al., 2024; Dou et al., 2024). Due to the difficulty of obtaining feedback in some domains, researchers are exploring learning without external feedback (Tang et al., 2025; Zhou et al., 2025). Another axis of differentiation is whether feedback is based on the final output of the model (outcome-based) (Xin et al., 2024; Ankner et al., 2024; DeepSeek-AI et al., 2025) or its intermediate steps (process-based) (Uesato et al., 2022; Lightman et al., 2023; Wang et al., 2023; Lai et al., 2024; Lin et al., 2025). The humanline paradigm works with different forms of feedback and is independent of whether that feedback is outcome- or process-based.

**Prospect Theory**   Having revolutionized behavioral economics, prospect theory (Kahneman & Tversky, 1979; Tversky & Kahneman, 1992) has recently been incorporated into LLM alignment via Kahneman-Tversky Optimization (KTO) (Ethayarajh et al., 2024). Previously, it has had only a limited impact in machine learning, mostly in human-robot interaction research (Kwon et al., 2020; Sun et al., 2019; Chan et al., 2021). While KTO focuses on human biases in the value function (2), the humanline design pattern does so for the weighting function (4).

## B PROOFS

**Proposition 3.4 (restated)** For any input $x$ and bounded value function $v$, let the outcome of an output $y$ be its surprisal $\log[\pi_\theta(y|x)/\pi_{\text{ref}}(y|x)]$ and $Q$ be a candidate distribution over outcomes. Then to guarantee $|u(Z;\omega) - u(Z;Q)| \leq \delta$ for some $\delta \geq 0$, it suffices that $\sqrt{\text{KL}(\omega\|Q)} \leq \delta/(\sqrt{2}\|v\|_\infty)$.

*Proof.* Let $z_{x,y}$ denote the outcome of an input-output pair $(x,y)$, where as in Ethayarajh et al. (2024), it is measured as the surprisal term $\log[\pi_\theta(y|x)/\pi_{\text{ref}}(y|x)]$. Assume that the human value function is bounded (as is the case in prospect theory), that the human-perceived distribution has support subsuming that of $Q$ (i.e., $\text{supp}(Q) \subseteq \text{supp}(\omega)$), and that $z_{x,y}$ is measurable with respect to the support of both distributions. Note that $\omega(z_{x,y})$ denotes the subjective probability (weight) assigned to output $y$ based on its outcome $z_{x,y}$, not a probability distribution over $y$ itself. That is, $\omega$ is a distortion of the cumulative distribution over outcomes (surprisals), as defined in Eq. (4). Then using Definition (3.3) of subjective utility:

$$
\begin{aligned}
|u(Z;\omega) - u(Z;Q)| &= \left| \sum_y \omega(z_{x,y})v(z_{x,y}) - \sum_y Q(z_{x,y})v(z_{x,y}) \right| \\
&= \left| \sum_y (\omega(z_{x,y}) - Q(z_{x,y}))v(z_{x,y}) \right| \\
&\leq \sum_y |v(z_{x,y})| \, |\omega(z_{x,y}) - Q(z_{x,y})| \quad \text{(triangle inequality)} \\
&\leq \|v\|_\infty \|\omega - Q\|_1 \\
&\leq \|v\|_\infty \sqrt{2 \cdot \text{KL}(\omega\|Q)} \quad \text{(Pinsker's inequality)}
\end{aligned}
$$

Then if $\sqrt{\text{KL}(\omega\|Q)} \leq \delta/(\sqrt{2}\|v\|_\infty)$, we get $|u(Z;\omega) - u(Z;Q)| \leq \delta$. $\qquad\square$

**Proposition 4.1 (restated)** Under typical conditions, for any context $x$, simulating output sequences $y$ from $\omega$ is equivalent to performing token-wise rejection sampling with the rejection criterion

$$\pi_\theta(y_t|x; y_{<t})/\pi_{\text{ref}}(y_t|x; y_{<t}) < M'_\theta B$$

where $B \sim \text{Beta}(\gamma, 1)$, $M'_\theta$ is a finite upper bound on the LHS for all tokens in the vocabulary, and $\gamma \in (0, 1]$ is the capacity function constant.

*Proof.* We consider the following conditions:

1. The proposal distribution is the current iteration of the reference model: i.e., any output sequence was produced by autoregressively sampling tokens from $\pi_{\text{ref}}(\cdot|x, y_{i,<t})$. Moreover, $\pi_{\text{ref}}(\cdot|x, y_{i,<t})$ and $\pi_\theta(\cdot|x, y_{i,<t})$ have the same support and a finite likelihood ratio bound. In practice, this means that they share a vocabulary (which holds because they are identical at $t = 0$); and that each possible token has non-zero probability, which arises trivially from a softmax output distribution.

2. The capacity functions $\Omega^+, \Omega^-$ have the standard functional form (3), implying that the human-biased CDF has the same structural form for all outcomes. This was found to hold in Tversky & Kahneman (1992) (for probability in a more general sense).

3. For context $x$ with output sequence $y_i$, let surprisal $z_i = \log[\pi_\theta(y|x)/\pi_{\text{ref}}(y|x)]$ denote the outcome. The cumulative probability of outcomes with higher absolute surprisal than $z_i$ is negligible (i.e., the vast majority of all possible output sequences is nonsensical or irrelevant, which holds trivially). To be more specific, for any given prompt, the possible output space of $n$-length sequences is very large. Only a small minority of possible completions are good or bad enough that we want to explicitly align towards or away from them, and the cumulative probability mass of better or worse completions is negligible.

Under these conditions, given outcome $z_i$, $\sum_{j>i} p_j \approx 0$ for $\Omega^+$ and $\sum_{j<i} p_j \approx 0$ for $\Omega^-$. Following from (4), the weight (i.e., subjective probability) of a sequence is

$$\omega(z) \approx \frac{p^\gamma}{(p^\gamma + (1-p)^\gamma)^{1/\gamma}}$$

For a sufficiently long sequence, the denominator will approach 1, meaning $\omega(z) \approx p^\gamma$. The numerator (and thus the weight) can be factorized over tokens as $p^\gamma = \prod_t p_t^\gamma$, meaning that instead of rejection-sampling entire sequences, we can just rejection sample one token at time. Because the policy and reference models yield softmax distributions over a finite shared vocabulary, for any given $(x, y_{<t})$ the likelihood ratio for a fixed (policy, reference) pair takes finitely many positive values, so there exists a finite bound $M'_\theta$. Our results only require that this maximum exists; they do not require $M'_\theta$ to be small, known, or independent of vocabulary size. Note that our results only need a finite bound for each fixed $\theta$, not a uniform covering argument over the entire parameter space. Then:

$$\frac{\pi_\theta^\gamma(y_t|x; y_{<t})}{\pi_{\text{ref}}^\gamma(y_t|x; y_{<t})} < M_\theta \cdot U \iff \frac{\pi_\theta(y_t|x; y_{<t})}{\pi_{\text{ref}}(y_t|x; y_{<t})} < M_\theta^{\frac{1}{\gamma}} \cdot U^{\frac{1}{\gamma}}$$

where $U \sim \text{Uniform}(0, 1)$.

Let $B \triangleq U^{\frac{1}{\gamma}}$, where $\gamma \in (0, 1]$ is the capacity function constant in (3). To get the density of this new random variable, we apply the transformation rule, noting that because $U$ is uniform on $[0, 1]$, $f_U(\cdot) = 1$:

$$f_B(b) = f_U(b^\gamma) \left| \frac{d}{db} b^\gamma \right| = 1 \cdot \gamma b^{\gamma-1}$$

This is the density of $\text{Beta}(\gamma, 1)$. Therefore,

$$B \sim \text{Beta}(\gamma, 1), \quad M'_\theta = M_\theta^{\frac{1}{\gamma}}.$$

$\square$

**Theorem 4.3 (restated)** The clipped component in PPO/GRPO is a special case of humanline sampling that arises under limit conditions.

*Proof.* Let $B_P$ denote the Beta random variable in Definition 4.2 and $M_P$ its corresponding constant that bounds the likelihood ratio. By definition, its mean and variance are

$$\mathbb{E}[B_P] = \frac{\gamma_P}{\gamma_P + \beta_P}, \qquad \text{Var}[B_P] = \frac{\gamma_P \beta_P}{(\gamma_P + \beta_P)^2(\gamma_P + \beta_P + 1)}.$$

Let $k, \epsilon_P \in \mathbb{R}^+$ be constants such that $\epsilon_P < M_P$. Setting $\gamma_P = \frac{k\epsilon_P}{M_P}$, $\beta_P = k(1 - \frac{\epsilon_P}{M_P})$, we get

$$\mathbb{E}[B_P] = \frac{k\frac{\varepsilon_P}{M_P}}{k\frac{\varepsilon_P}{M_P} + k(1 - \frac{\varepsilon_P}{M_P})} = \frac{\varepsilon_P}{M_P},$$

$$\text{Var}[B_P] = \frac{k\frac{\varepsilon_P}{M_P} k(1 - \frac{\varepsilon_P}{M_P})}{(k\frac{\varepsilon_P}{M_P} + k(1 - \frac{\varepsilon_P}{M_P}))^2(k\frac{\varepsilon_P}{M_P} + k(1 - \frac{\varepsilon_P}{M_P}) + 1)} = \frac{\frac{\varepsilon_P}{M_P}\left(1 - \frac{\varepsilon_P}{M_P}\right)}{k+1}.$$

As $k \to \infty$, $\forall \delta > 0$, $\text{Pr}(|B_P - \epsilon_P/M_P| \geq \delta) \to 0$ (i.e., we deterministically sample the mean).

Similarly, for $B_R, M_R$ in Definition 4.2, let $\epsilon_R \in \mathbb{R}^+$ be such that $\epsilon_R > 1/M_R$ and set $\gamma_R = k/(\epsilon_R M_R), \beta_R = k(1 - 1/(\epsilon_R M_R))$. Then as $k \to \infty$, $\forall \delta > 0$, $\text{Pr}(|B_R - 1/\epsilon_R M_R| \geq \delta) \to 0$.

Thus as $k \to \infty$, the rejection criteria in token-level humanline sampling simplify to:

$$\frac{\pi_\theta(y_t|x; y_{<t})}{\pi_{\text{ref}}(y_t|x; y_{<t})} < M_P \cdot \frac{\varepsilon_P}{M_P} = \varepsilon_P, \qquad \frac{\pi_{\text{ref}}(y_t|x; y_{<t})}{\pi_\theta(y_t|x; y_{<t})} < M_R \cdot \frac{1}{\varepsilon_R M_R} = \frac{1}{\varepsilon_R}$$

which means that the tokens that are accepted satisfy:

$$\varepsilon_P \leq \frac{\pi_\theta(y_t|x; y_{<t})}{\pi_{\text{ref}}(y_t|x; y_{<t})} \leq \varepsilon_R.$$

Recall that $M_P, M_R$ are upper bounds on the likelihood ratios, and given that $\pi_\theta, \pi_{\text{ref}}$ are distributions over the same support and are generally not identical, there will exist tokens for which these ratios are both greater than 1. Thus $M_P \geq 1$ and $M_R \geq 1$. For any fixed $\varepsilon \in (0, 1)$, it is therefore a given that $(1 - \epsilon) < M_P$ and $(1 + \epsilon) > 1/M_R$, meeting the constraints imposed earlier on $\epsilon_P, \epsilon_R$. Letting $\varepsilon_R = 1 + \varepsilon$ and $\varepsilon_P = 1 - \varepsilon$, we get the following inequality that is satisfied by accepted tokens:

$$1 - \varepsilon \leq \frac{\pi_\theta(y_t|x; y_{<t})}{\pi_{\text{ref}}(y_t|x; y_{<t})} \leq 1 + \varepsilon.$$

This recovers the clipped term in PPO and GRPO, where the ratio $\pi_\theta(y_t|x; y_{<t})/\pi_{\text{ref}}(y_t|x; y_{<t})$ for each token is clipped to the range $[1 - \epsilon, 1 + \epsilon]$. Note that the equivalence is not only due to the likelihood ratios being bound to the same range, but also due to ratios outside the range contributing nothing to the gradient, either due to the shape of the clipping function (in PPO/GRPO) or due to being explicitly detached from the computation graph (in humanline sampling). □

## C  ALGORITHMS

### C.1  DEFINITIONS

#### C.1.1  OFFLINE DPO/KTO

DPO (Rafailov et al., 2023) and KTO (Ethayarajh et al., 2024) were originally proposed as offline algorithms, and we use the original definitions without any change for offline DPO/KTO. Where $(x, y_w, y_l)$ is a tuple from an offline dataset $\mathcal{D}$ representing a preference for output $y_w$ over $y_l$ given context $x$, the DPO loss is:

$$L_{\text{DPO}}(\pi_\theta, \pi_{\text{ref}}) = \mathbb{E}_{x, y_w, y_l \sim D}\left[-\log \sigma\left(\beta \log \frac{\pi_\theta(y_w|x)}{\pi_{\text{ref}}(y_w|x)} - \beta \log \frac{\pi_\theta(y_l|x)}{\pi_{\text{ref}}(y_l|x)}\right)\right] \tag{6}$$

where $\beta \in \mathbb{R}^+$ is a hyperparameter and $\sigma$ is the sigmoid function.

Instead of paired preferences, KTO frames outputs $y$ as undesirable or desirable. Where $\lambda_y \in \mathbb{R}^+$ denotes $\lambda_D(\lambda_U)$ when $y$ is desirable(undesirable) respectively, the default KTO loss is:

$$L_{\text{KTO}}(\pi_\theta, \pi_{\text{ref}}) = \mathbb{E}_{x, y \sim D}[\lambda_y - v(x, y)] \tag{7}$$

where

$$r_\theta(x, y) = \log \frac{\pi_\theta(y|x)}{\pi_{\text{ref}}(y|x)}$$

$$z_0 = \text{KL}(\pi_\theta(y'|x)\|\pi_{\text{ref}}(y'|x))$$

$$v(x, y) = \begin{cases} \lambda_D \sigma(\beta(r_\theta(x, y) - z_0)) & \text{if } y \sim y_{\text{desirable}}|x \\ \lambda_U \sigma(\beta(z_0 - r_\theta(x, y))) & \text{if } y \sim y_{\text{undesirable}}|x \end{cases}$$

There is no backpropagation through $z_0$; it exists purely to control the loss saturation. In practice, for the sake of efficiency, a shared KL estimate is used for all examples in the same batch by taking the average $r_\theta$ over mismatched input-output pairs $(x, y')$. In our experiments, for an apples-to-apples comparison across methods, we break up DPO preference pairs to get unpaired data for KTO, although we use twice the batch size so that the same number of steps are taken.

#### C.1.2  ONLINE DPO/KTO

Our implementation of online DPO combines features of the online DPO implementation in Guo et al. (2024), iterative DPO in Xu et al. (2024b), and semi-online DPO in Lanchantin et al. (2025a). Like Guo et al. (2024) and Lanchantin et al. (2025a), we sample completions from the policy being actively aligned. However, like Lanchantin et al. (2025a), we do not sample every step, because it is slower, more computationally expensive, and leads to worse results. Asynchronous training is typical in RLHF, especially in large-scale distributed settings (Noukhovitch et al., 2024).

We find that sampling once every 1024 contexts (i.e., 32 steps) leads to best performance in the instruction-following setting, which we call one *round*. For each of the 1024 contexts in a round, we sample 8 completions $\{y_i\}_{i=1}^8 \sim \pi_\theta(\cdot|x)$, score them with a reward model, compare the highest- and lowest-scoring $y$ for each $x$, and construct a paired preference $(x, y_w, y_l)$ if the difference in score exceeds $\tau = 0.01$. We use this methodology because it is nearly identical to how the offline instruction-following data was constructed (Meng et al., 2024), even using the exact same reward model (Wang et al., 2024) and contexts $x$ (Cui et al., 2024) to enable an apples-to-apples comparison. The differences are: (1) our use of threshold $\tau$, which is needed to construct feedback that is sufficiently discriminative; (2) using 22% more contexts than in the offline data to adjust for the fact that using $\tau$ leads to roughly 18% of the preferences (albeit low-signal preferences) being discarded. Therefore the volume of data seen by both offline DPO and online DPO is approximately the same, although the latter sees more diversity in contexts, which may provide an additional advantage. At the end of a round, the policy is checkpointed, and at the start of the next round, the new policy and reference model are loaded from this checkpoint.

Online KTO is implemented the exact same way, albeit the final loss is calculated with (7) instead of (6). Notably, we construct DPO-style paired preferences before breaking them up to create unpaired data for KTO, instead of directly creating unpaired data using positive and negative thresholds. Not only does this allow for a better comparison with DPO, but it also works better in practice. See Algorithm 1 for the pseudo-code of Online DPO/KTO.

---

**Algorithm 1** Online DPO / KTO

---

**Input:** initial policy model $\pi_{\theta_{\text{init}}}$; reward model $r_\varphi$; reward threshold $\tau$; prompts $\mathcal{D}$; hyperparameters $\beta, \lambda_{\text{desirable}}, \lambda_{\text{undesirable}}$
**Output:** policy model $\pi_\theta$

 1: Initiate policy model $\pi_\theta \leftarrow \pi_{\theta_{\text{init}}}$
 2: **for** round $= 1$ **to** $N$ **do**
 3:     Set reference model $\pi_{\text{ref}} \leftarrow \pi_\theta$;  $\mathcal{D}_{\text{train}} \leftarrow \varnothing$
 4:     Sample a batch of contexts $\mathcal{D}_b$ from $\mathcal{D}$, where $|\mathcal{D}_b| = 1024$
 5:     **for** prompt $x \in \mathcal{D}_b$ **do**                                ▷ Online sampling and relabeling
 6:         Sample $G$ outputs $\{y_i\}_{i=1}^G \sim \pi_\theta(\cdot \mid x)$, where $G = 8$
 7:         Compute rewards $\{r_i\}_{i=1}^G$ for each $y_i$ via $r_\varphi$
 8:         **if** $|\max_j r_j - \min_j r_j| \geq \tau$ **then**             ▷ ~18% samples will be filtered out
 9:             $(y_w, y_l) \leftarrow (\arg\max_j r_j, \arg\min_j r_j)$
10:             $\mathcal{D}_{\text{train}} \leftarrow \mathcal{D}_{\text{train}} \cup \{(x, y_w, y_l)\}$
11:         **end if**
12:     **end for**
13:     **for** batch $\mathcal{B} \in \mathcal{D}_{\text{train}}$ **do**                ▷ Train with newly generated preference data
14:         Compute token-level surprisal $\hat{r}_{i,t}$ for every token $t$ in $(y_w, y_l) \sim \mathcal{B}$ via $\pi_\theta$ and $\pi_{\text{ref}}$
15:         Update $\pi_\theta$ by maximizing the DPO / KTO objective (Eq. 6; 7)
16:     **end for**
17: **end for**

---

**Algorithm 2** Offline GRPO

---

**Input:** initial policy model $\pi_{\theta_{\text{init}}}$; prompts and completions $\mathcal{D}$; hyperparameters $\beta, \epsilon$
**Output:** policy model $\pi_\theta$

 1: Initiate policy model $\pi_\theta \leftarrow \pi_{\theta_{\text{init}}}$; reference model $\pi_{\text{ref}} \leftarrow \pi_\theta$
 2: **for** step $= 1, ..., \text{M}$ **do**
 3:     Sample a batch $\mathcal{D}_b$ from $\mathcal{D}$                   ▷ Train with off-policy preference data
 4:     **for** prompt $x \in \mathcal{D}_b$ **do**
 5:         Set $G \leftarrow \{y_w, y_l\}$ from off-policy $(x, y_w, y_l)$ tuples      ▷ Default group size = 2
 6:         Compute token-level surprisal $\hat{r}_{i,t}$ for every token $t$ in $\{y_w, y_l\}$ via $\pi_\theta$ and $\pi_{\text{ref}}$
 7:         Compute $\hat{A}_{i,t}$ for every token $t$ in $\{y_w, y_l\}$ through group relative advantage estimation
 8:         Update $\pi_\theta$ by maximizing the GRPO objective (Eq. 1)
 9:     **end for**
10: **end for**

---

### C.1.3 OFFLINE GRPO

Given that GRPO is inherently an online method (Shao et al., 2024), we make a few different changes to create an offline variant, which largely follow those made by Ethayarajh et al. (2024) to make an offline variant of PPO. For one, instead of sampling new completions, we take tuples $(x, y_w, y_l)$ in an offline preference dataset (e.g., UltraFeedback (Cui et al., 2024)) and treat them as a group of two: $G = \{y_i\}_{i=1}^2 = \{y_w, y_l\}$. The reference model is never updated: its weights remain those of the policy at initialization. See Algorithm 2 for the pseudo-code of offline GRPO.

### C.1.4 ONLINE GRPO

Instead of sampling every step, we sample data as in Online DPO/KTO, the only difference being that we retain the raw scores from the scoring step so that they can later be fed into the loss calculation (1). We use this approach to allow for an apples-to-apples comparison with Online DPO/KTO, as well as because some asynchronicity is usually permitted in practice and we find that sampling once per round (i.e., roughly every 32 steps) does not degrade performance. A consequence of this choice is that the group size is exactly 2 for all contexts, making the relative advantages either -1 or 1. Naturally, this does not unlock the full potential of GRPO, since one of its strengths is its ability to leverage scalar rewards. However, we consider it more important that the volume of training data to be roughly the same across different variants and methods. We also use DAPO-style normalization

(i.e., taking the average loss over the number of tokens in the batch instead of within a sequence), as we find this leads to better performance on instruction-following (Yu et al., 2025). Lastly, we reuse the reference model as the base model for calculating the KL penalty, both because it saves us the space of storing a third model and because prior work has identified the KL penalty to not be of much import, allowing its estimate to be less precise.

## C.2 HUMANLINE IMPLEMENTATION

The instruction-following experiments were done in a fork of the HALOs repository. Below, we provide a relatively straightforward implementation of the humanline design pattern for DPO, KTO and GRPO, which is triggered by setting `self.config.humanline = True` in our codebase.

The mathematical reasoning experiments were implemented in a fork of the Open-R1 repository (Hugging Face, 2025), which itself is based on Huggingface's TRL library. Because of this, humanline syncing is implemented differently, by over-writing callback methods: before the optimizer step happens, the current state of the policy is stored locally; at the end of the step, the stored policy is loaded into the reference model. This improves stability when doing distributed training with ZeRO2 (Rajbhandari et al., 2020).

### C.2.1 HUMANLINE SYNCING

In our codebase, we first modify the training loop to implement humanline syncing. For the sake of brevity, we highlight only the relevant changes in `train()` and omit code used for logging. The `accelerator` object is used to manage distributed training with FSDP in our own codebase:

```python
def train():
    ...
    self.optimizer.zero_grad()
    loss, metrics = self.get_batch_metrics(batch)
    self.accelerator.backward(loss)
    grad_norm = self.accelerator.clip_grad_norm_(self.policy.parameters(),
        self.config.model.max_grad_norm)

    if self.config.loss.sync_reference or self.config.humanline:
        self.sync_reference_with_policy()

    self.optimizer.step()
    self.scheduler.step()
    ...

def sync_reference_with_policy(self):
    """
    Update the reference model to have the policy weights.
    """
    if self.batch_counter % self.config.sync_freq == 0:
        state_dict = self.accelerator.unwrap_model(self.policy).state_dict()
        self.accelerator.unwrap_model(self.reference_model).load_state_dict(state_dict)
        self.accelerator.wait_for_everyone()
```

Humanline clipping is even easier to implement, although it has to be implemented in two different places in our codebase because of the different abstractions used for DPO/KTO and GRPO:

```python
def get_sequence_rewards(self,
    policy_logps: torch.FloatTensor,
    reference_logps: torch.FloatTensor,
    length_normalized=False,
    ):
    """
    If regular alignment, return the surprisal for the sequence
    (log [policy(y|x)/reference(y|x)]).
    This is called the "sequence reward", following DPO terminology.
    Apply humanline if specified.
```

```
12      Args:
13          policy_logps: token-level probabilities according to policy
14              (microbatch_size, maximum sequence length)
15          reference_logps: token-level probabilities according to reference
16              model (microbatch_size, maximum sequence length)
17          length_normalized: divide the sequence reward by the number of
18              non-rejected tokens
19
20      Returns:
21          The sequence-level rewards (microbatch_size, 1).
22      """
23      if self.config.humanline:
24          token_rewards = (policy_logps - reference_logps).clamp(
25              self.config.log_epsilon_P, self.config.log_epsilon_R)
26      else:
27          token_rewards = policy_logps - reference_logps
28
29      normalization_factor = (token_rewards.abs() != 0).float().sum(-1) \
30          if length_normalized else 1
31      sequence_rewards = token_rewards.sum(-1) / normalization_factor
32
33      return sequence_rewards
34
35
36  class DPOTrainer(PairedPreferenceTrainer):
37      def loss(self,
38          batch: Dict,
39          policy_chosen_logps: torch.FloatTensor,
40          policy_rejected_logps: torch.FloatTensor,
41          reference_chosen_logps: torch.FloatTensor,
42          reference_rejected_logps: torch.FloatTensor,
43          *args,
44          ):
45          """Compute the DPO loss for a batch of policy and reference model
46          token-level log probabilities."""
47
48          # apply humanline clipping via get_sequence_rewards on token-level
49          # log probabilities before they are fed into loss computation
50          chosen_rewards = self.get_sequence_rewards(policy_chosen_logps,
51              reference_chosen_logps)
52          rejected_rewards = self.get_sequence_rewards(policy_rejected_logps,
53              reference_rejected_logps)
54
55          chosen_rewards *= self.config.loss.beta
56          rejected_rewards *= self.config.loss.beta
57
58          losses = -F.logsigmoid(chosen_rewards - rejected_rewards)
59
60          return losses, chosen_rewards.detach(), rejected_rewards.detach()
61
62
63  class KTOTrainer(UnpairedPreferenceTrainer):
64      def loss(self,
65          batch: Dict,
66          policy_chosen_logps: torch.FloatTensor,
67          policy_rejected_logps: torch.FloatTensor,
68          policy_KL_logps: torch.FloatTensor,
69          reference_chosen_logps: torch.FloatTensor,
70          reference_rejected_logps: torch.FloatTensor,
71          reference_KL_logps: torch.FloatTensor,
72          *args,
73          ):
74          """Compute the KTO loss for a batch of policy and
75          reference model log probabilities.
76
```

```
77            If generation y ~ p_desirable, we have the 'desirable' loss:
78                L(x, y) := 1 - sigmoid(beta * ([log p_policy(y|x)
79                    - log p_reference(y|x)] - KL(p_policy || p_reference)))
80            If generation y ~ p_undesirable, we have the 'undesirable' loss:
81                L(x, y) := 1 - sigmoid(beta * (KL(p_policy || p_reference)
82                    - [log p_policy(y|x) - log p_reference(y|x)]))
83
84            The desirable losses are weighed by config.loss.desirable_weight.
85            The undesirable losses are weighed by config.loss.undesirable_weight.
86            This should be used to address imbalances in the ratio of
87                desirable:undesirable examples respectively.
88            The KL term is estimated by matching x with unrelated outputs y',
89                then calculating the average log ratio
90                log p_policy(y'|x) - log p_reference(y'|x).
91            """
92            if policy_chosen_logps.shape[0] != 0:
93                chosen_rewards = self.get_sequence_rewards(
94                    policy_chosen_logps, reference_chosen_logps)
95            else:
96                chosen_rewards = torch.Tensor([]).to(self.policy_dtype).to(
97                    self.accelerator.device)
98
99            if policy_rejected_logps.shape[0] != 0:
100               rejected_rewards = self.get_sequence_rewards(
101                   policy_rejected_logps, reference_rejected_logps)
102           else:
103               rejected_rewards = torch.Tensor([]).to(self.policy_dtype).to(
104                   self.accelerator.device)
105
106           # For KTO, humanline also applies to the KL term
107           KL_rewards = self.get_sequence_rewards(policy_KL_logps.detach(),
108               reference_KL_logps.detach())
109           KL = (KL_rewards.sum() / (KL_rewards.abs() != 0).float().sum().item()
110               .clamp(min=0)
111
112           if policy_chosen_logps.shape[0] != 0:
113               chosen_losses = self.config.loss.desirable_weight *
114                   (1 - F.sigmoid(self.config.loss.beta * (chosen_rewards - KL)))
115           else:
116               chosen_losses = torch.Tensor([]).to(self.policy_dtype).to(
117                   self.accelerator.device)
118
119           if policy_rejected_logps.shape[0] != 0:
120               rejected_losses = self.config.loss.undesirable_weight *
121                   (1 - F.sigmoid(self.config.loss.beta * (KL - rejected_rewards)))
122           else:
123               rejected_losses = torch.Tensor([]).to(self.policy_dtype).to(
124                   self.accelerator.device)
125
126           losses = torch.cat((chosen_losses, rejected_losses), 0)
127
128           return losses, chosen_rewards.detach(), rejected_rewards.detach(),
129               KL.detach()
```

For both DPO and KTO, we apply the same token-wise likelihood clipping with the function get_sequence_rewards as shown above. For GRPO, we do the same but with the function get_ratios to return the probability ratio under the policy and the reference models instead of the log probability ratio, as defined in the clipped surrogate objective. Note that we clamp in log-space for greater numerical precision.

```
1   def get_ratios(self,
2       policy_logps,
3       reference_logps,
4       ):
```

```python
5          """
6          If regular alignment, return the token-level probability ratio
7              under the policy vs the reference [policy(y|x)/reference(y|x)].
8          Apply humanline if specified.
9
10         Args:
11             policy_logps: token-level probabilities according to policy
12                 (microbatch_size, maximum sequence length)
13             reference_logps: token-level probabilities according to
14                 reference model (microbatch_size, maximum sequence length)
15
16         Returns:
17             The probability ratios (microbatch_size, sequence length) if
18                 sequence_level; otherwise, (microbatch_size, 1)
19         """
20         if self.config.humanline:
21             logratio = (policy_logps - reference_logps).clamp(
22                 self.config.log_epsilon_P, self.config.log_epsilon_R
23             )
24         else:
25             logratio = policy_logps - reference_logps
26
27         ratio = logratio.exp()
28
29         return ratio
30
31
32 class GRPOTrainer(BasicTrainer):
33     def loss(self,
34         batch: Dict,
35         policy_logps: torch.FloatTensor,
36         reference_logps: torch.FloatTensor,
37         advantages: torch.FloatTensor,
38         group_size: torch.FloatTensor,
39         *args,
40         ):
41         """
42         Compute the GRPO loss.
43
44         Args:
45             policy_logps: log probability of the output under the policy
46                 (microbatch_size, sequence_length)
47             reference_logps: log probability of the output under the
48                 reference model (microbatch_size, sequence_length)
49             advantages: sequence level advantages (microbatch_size,)
50             group_size: number of outputs (in entire batch) belonging to
51                 prompt associated with sequence (microbatch_size,)
52
53         Returns:
54             average loss, average KL, average weighted advantage,
55             average unweighted advantage
56         """
57         # apply humanline clipping via get_ratios on token-level
58         # log probabilities which returns probability ratios
59         ratio = self.get_ratios(policy_logps, reference_logps)
60         masks = (batch['target_labels'][:, 1:] != -100).clone().to(
61             self.policy_dtype)
62
63         advantages = advantages.unsqueeze(-1)
64         group_size = group_size.unsqueeze(-1)
65
66         weighted_adv = advantages * ratio
67         # probability ratios get clipped again in the GRPO surrogate
68         # objective controlled by a separate hyperparameter epsilon
69         weighted_adv_clipped = advantages * ratio.clamp(
```

```
70                1 - self.config.loss.epsilon,
71                1 + self.config.loss.epsilon)
72
73            # humanline clipping does not apply to KL term in GRPO
74            per_token_KL = torch.exp(reference_logps - policy_logps)
75                - (reference_logps - policy_logps) - 1
76            per_token_loss = -torch.min(weighted_adv, weighted_adv_clipped)
77                + self.config.loss.beta * per_token_KL
78
79            # do DAPO-style normalization
80            return masked_mean(per_token_loss, masks),
81                masked_mean(per_token_KL.detach(), masks),
82                masked_mean(weighted_adv.abs().detach(), masks),
83                advantages.abs().mean()
84
```

# D  ADDITIONAL EXPERIMENTS

Table 1: Hyperparameters that are common to all of our instruction-following experiments, across different alignment objectives and different variants for each objective. Note that we generate much more data than we ultimately use during online training (see Appendix C.1 for details). Note the data volume for KTO/GRPO is twice that of DPO because the latter operates on paired preferences containing two sequences each.

| Hyperparameter | Value |
|---|---|
| Optimizer | AdamW |
| AdamW $\beta_1$ | 0.9 |
| AdamW $\beta_2$ | 0.999 |
| AdamW $\epsilon$ | 1e-5 |
| Weight Decay | 1e-2 |
| Warmup | 10% |
| Offline Training Examples | 10K (DPO) / 20K (KTO, GRPO) |
| Offline Batch Size | 32 (DPO) / 64 (KTO, GRPO) |
| Online Training Contexts | 12288 |
| Online Batch Size | 32 (DPO) / 64 (KTO, GRPO) |
| Round (Number of Contexts) | 1024 |
| Generations per Context (sampled) | 8 |
| Generations per Context (after filtering) | 2 or 0 |
| Maximum Generation Length | 2048 |
| Top-$p$ (Nucleus Sampling) | 0.95 |
| Sampling Temperature | 0.7 |
| Reward Threshold $\tau$ | 0.01 |
| Humanline $\log \epsilon_P$ | $-1.5$ |
| Humanline $\log \epsilon_R$ | 1.5 |
| Humanline $k$ | 1 |

Table 2: Hyperparameters that are common to all of our mathematical reasoning experiments with GRPO. We use the setup in Huggingface's `Open-R1` (Hugging Face, 2025), except instead of placing equal weight on the `tag_count`, `format`, and `accuracy` rewards, we place weights of 1, 1, and 8 respectively (i.e., emphasizing accuracy over the rest). Doing humanline syncing every step ($k = 1$) will lead to collapse in this setup because of the smaller models involved; $k \in [12, 20]$ closes the gap with the online reward curves within 1000 steps.

| Hyperparameter | Value |
|---|---|
| Optimizer | AdamW |
| AdamW $\beta_1$ | 0.9 |
| AdamW $\beta_2$ | 0.999 |
| AdamW $\epsilon$ | 1e-8 |
| Weight Decay | 0 |
| Warmup | 10% |
| Learning Rate | 1e-6 |
| Max Gradient Norm | 1.0 |
| Training Contexts | 12K |
| Batch Size (incl. duplicate prompts due to groups) | 256 |
| Group Size | 8 |
| Maximum Generation Length | 2048 |
| Top-$p$ (Nucleus Sampling) | 1.0 |
| Sampling Temperature | 0.7 |
| Humanline $\log \epsilon_P$ | $-1.5$ |
| Humanline $\log \epsilon_R$ | 1.5 |
| Humanline $k$ | $[12, 24]$ |

Table 3: The performance of `Llama3-8B-Instruct` trained with all variants of all objectives on AlpacaEval2, along with the objective-specific hyperparameters. Note that while humanline alignment usually reduces the average length (**Length**) of completions, this is not a universal characteristic of humanline variants, but of the data and hyperparameters used; results from training on a different version of UltraFeedback, where completions are sampled from a different model, lead to offline+humanline variants having roughly the same length as their offline counterparts (Table 4). The hyperparameters that require the most adjusting are the learning rate (**LR**) and the maximum gradient clipping norm (**Max Norm**); going from offline to online requires the LR and Max Norm to be scaled by 0.5x-1x, but adding a humanline variant on top can increase or decrease the Max Norm (see §5.2.1 for a discussion of why). Objective-specific hyperparameters remain fixed across variants to allow for a fair comparison, with the exception of Offline GRPO $\epsilon$, which needs to be much larger when the reference model is fixed.

| Objective | LR | $\beta$ | $\lambda_D$ | $\lambda_U$ | Max Norm | LC-WR ↑ | WR ↑ | Length |
|---|---|---|---|---|---|---|---|---|
| Offline KTO | 5.0e-6 | 0.25 | 1.1 | 1 | 1.0 | 17.40 | 14.13 | 1658 |
| Offline+Humanline KTO | 5.0e-6 | 0.25 | 1.1 | 1 | 4.0 | 22.19 | 14.70 | 1407 |
| Online KTO | 2.5e-6 | 0.25 | 1.1 | 1 | 0.5 | 22.45 | 19.47 | 1744 |
| Online+Humanline KTO | 2.5e-6 | 0.25 | 1.1 | 1 | 0.1 | **22.79** | 18.78 | 1663 |

| Objective | | LR | $\beta$ | | Max Norm | LC-WR ↑ | WR ↑ | Length |
|---|---|---|---|---|---|---|---|---|
| Offline DPO | | 5.0e-6 | 0.10 | | 1.0 | 18.07 | 16.07 | 1767 |
| Offline+Humanline DPO | | 5.0e-6 | 0.10 | | 1.0 | 24.82 | 20.18 | 1637 |
| Online DPO | | 2.5e-6 | 0.10 | | 0.5 | 24.96 | 22.99 | 1828 |
| Online+Humanline DPO | | 2.5e-6 | 0.10 | | 1.0 | **26.84** | 23.64 | 1774 |

| Objective | LR | $\beta$ | $\epsilon$ | | Max Norm | LC-WR ↑ | WR ↑ | Length |
|---|---|---|---|---|---|---|---|---|
| Offline GRPO | 5.0e-6 | 0.01 | 0.50 | | 1.0 | 15.52 | 12.61 | 1648 |
| Offline+Humanline GRPO | 5.0e-6 | 0.01 | 0.15 | | 1.0 | 25.24 | 18.11 | 1488 |
| Online GRPO | 5.0e-6 | 0.01 | 0.15 | | 1.0 | 25.05 | 18.82 | 1529 |
| Online+Humanline GRPO | 5.0e-6 | 0.01 | 0.15 | | 0.5 | **26.10** | 21.57 | 1647 |

Table 4: AlpacaEval2 results when `Llama3-8B-Instruct` is trained on two different versions of offline UltraFeedback ArmoRM (Meng et al., 2024), one where completions are generated by `Llama3-8B-Instruct` (a separate unaligned version producing offline data, not to be confused with the policy) and another where completions are generated by `Gemma2-9B-Instruct`. The contexts are the same in both cases. Significant differences ($p < 0.05$) are highlighted in red. Although the performance is not significantly different in most cases, when trained on the `Llama3` completions, the offline+humanline-aligned policies tend to produce shorter completions than their offline counterparts; when trained on the `Gemma2` completions, this is not necessarily the case. Using humanline variants does not permit one to ignore the data, as it will always make a difference in the quality of the aligned model.

| Objective | Llama3-8B-Instruct Data | | | Gemma2-9B-Instruct Data | | |
|---|---|---|---|---|---|---|
| | LC-WR ↑ | WR ↑ | Length | LC-WR ↑ | WR ↑ | Length |
| Offline KTO | 17.40 | 14.13 | 1658 | 18.10 | 15.28 | 1698 |
| Offline+Humanline KTO | 22.19 | 14.70 | 1407 | 22.18 | 20.30 | 1836 |
| Offline DPO | 18.07 | 16.07 | 1767 | 18.63 | 15.44 | 1690 |
| Offline+Humanline DPO | 24.82 | 20.18 | 1637 | 26.26 | 21.91 | 1642 |
| Offline GRPO | 15.52 | 12.61 | 1648 | 12.64 | 10.84 | 1696 |
| Offline+Humanline GRPO | 25.24 | 18.11 | 1488 | 24.24 | 18.99 | 1587 |

Table 5: AlpacaEval2 results when `Gemma2-27B-Instruct` is aligned with DPO on two different versions of offline UltraFeedback ArmoRM (Meng et al., 2024), one where completions are generated by `Llama3-8B-Instruct (L3-8B)` and another where completions are generated by `Gemma2-9B-Instruct (G2-9B)`. The contexts are the same in both cases. Note that using the offline+humanline variant is only able to match the performance of the online variant when the offline off-policy data comes from the latter of the two sources.

| Objective (DPO Variants) | LC-WR ↑ | WR ↑ | Std. Err | LR | $\beta$ |
|---|---|---|---|---|---|
| Baseline | 45.90 | 35.45 | 1.54 | | |
| Offline (`L3-8B` Completions) | 48.59 | 32.36 | 1.58 | 2.5e-6 | 0.3 |
| Offline+Humanline (`L3-8B` Completions) | 56.27 | 44.49 | 1.67 | 2.5e-6 | 0.3 |
| Offline (`G2-9B` Completions) | 56.58 | 45.17 | 1.68 | 2.5e-6 | 0.1 |
| Offline+Humanline (`G2-9B` Completions) | **67.45** | 61.37 | 1.64 | 2.5e-6 | 0.1 |
| Online | 66.49 | 74.22 | 1.48 | 2.5e-6 | 0.1 |

Table 6: AlpacaEval2 results when `Llama3-8B-Instruct` is aligned with humanline DPO with different choices of humanline clipping hyperparameters $\epsilon_P, \epsilon_R$. Humanline syncing is done every step ($k = 1$) and other hyperparameters are fixed. The performance of the aligned model is fairly robust to the choice of clipping values in both directions, with most length-controlled winrates (**LC-WR**) falling within the standard error of the highest one. However, the length of the outputs does grow monotonically as the clipping range gets looser.

| $\log \epsilon_P$ | $\log \epsilon_R$ | LC-WR ↑ | Std. Err | Length |
|---|---|---|---|---|
| -2.0 | 1.5 | 23.33 | 1.35 | 1663 |
| -1.5 | 1.5 | **24.82** | 1.36 | 1637 |
| -1.0 | 1.5 | 24.37 | 1.33 | 1588 |
| -1.0 | 2.0 | 22.55 | 1.31 | 1619 |
| -1.0 | 3.0 | 23.98 | 1.33 | 1636 |

Figure 8: When aligning `Llama3-8B-Instruct` with humanline GRPO, the performance on instruction-following—measured here as the length-controlled winrate against a `GPT-4-Turbo` baseline—is robust to the frequency of humanline syncing up to $k = 4$. Past that point, syncing less frequently leads to a log-linear decline in performance. In other setups not shown here, such as our mathematical reasoning experiments with `Qwen2.5-1.5B-Instruct`, syncing less frequently is not only beneficial but necessary, since anything less than $k = 12$ introduces too much instability and leads to reward collapse.

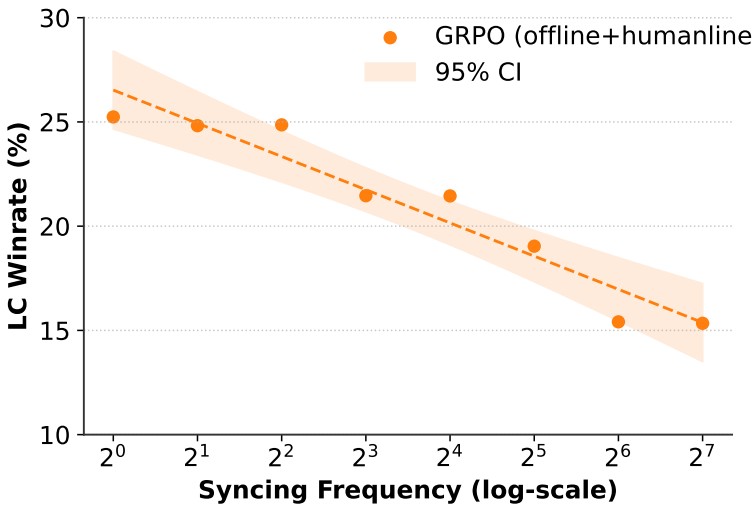

Figure 9: The performance benefits of the *humanline* variants of KTO/DPO/GRPO persist at larger scale with different model families, with `Gemma2-27B-Instruct` seeing a 1.15–1.30x improvement in performance. This is slightly smaller than the relative improvement seen by `Llama3-8B-Instruct`, and can be ascribed to the former being a better base model.

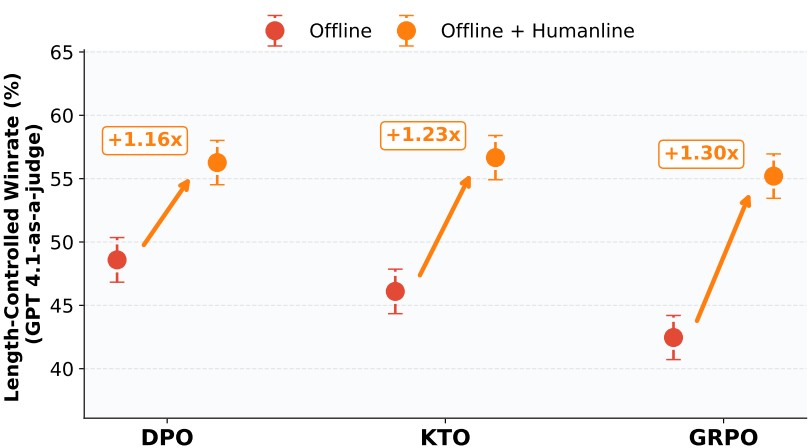

Figure 10: Trust region-style syncing (Gorbatovski et al., 2024) performs much worse than humanline syncing. In *offline+trust region*, we sync the reference model with the policy *after* the update every 1024 steps, the best performing setup both in Gorbatovski et al. (2024) and in our hyperparameter sweep. This suggests that it is not enough to merely sync the reference model; the way in which it is done matters as well.

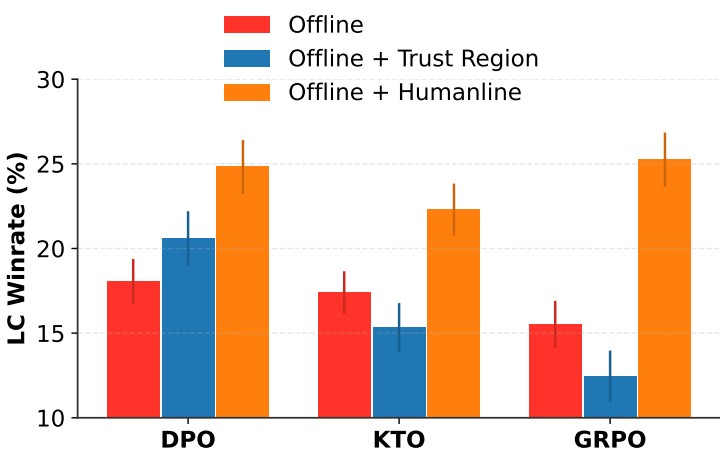

Figure 11: Average wall-clock time for aligning `Llama3-8B-Instruct` on UltraFeedback using FSDP across 4xH100 GPUs, reported with standard error and 95% confidence intervals across 5 random seeds. Note that offline+humanline GRPO takes almost twice as long as offline GRPO due to the syncing of the reference model weights. However, this is still less than 1/6 of the time needed to run online GRPO (without any overlapping of training/inference) while reaching the same performance (Figure 1).

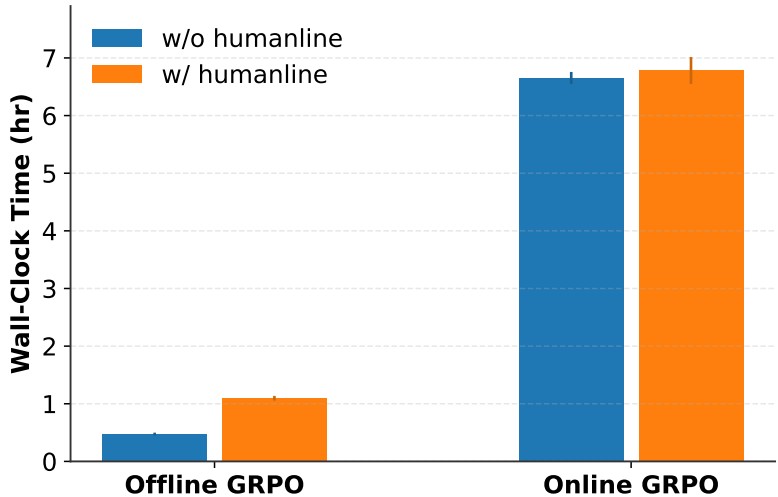

Table 7: Generations from the different DPO-aligned versions of `Gemma2-27B-Instruct` given a simple math question. Note that only the offline+humanline output explicitly states the weighting idea, explains why a simple sum/divide is only valid with equal weights, and then shows the per-quarter contributions—but without the verbosity of offline DPO.

| Prompt | In this math problem, we are asked to figure out the percentage of total profits that are made in each quarter. They are 90% in the first quarter, 92% in the second quarter, 81% in the third quarter and 88% in the fourth quarter. What is the total profit percent over a period of time, based upon these quarters? |
|---|---|
| **Baseline** | Here's how to calculate the average profit percentage over the year:
1. **Add up the profit percentages**: 90% + 92% + 81% + 88% = 351%
2. **Divide the total percentage by the number of quarters**: 351% / 4 = 87.75%
Therefore, the average profit percentage over the year is 87.75%. |
| **Offline** | Here's how to calculate the total profit percentage over the year:
**1. You can't directly add percentages like this.** Percentages represent parts of a whole, and simply adding them could result in a number over 100%, which doesn't make sense in this context.
**2. Think of it as weighted averages:** Each quarter's profit percentage reflects its contribution to the overall yearly profit. To find the total profit percentage, we need to calculate a weighted average.
**3. Here's the formula:**
\* **Weighted Average = (Q1% \* Q1 Weight) + (Q2% \* Q2 Weight) + (Q3% \* Q3 Weight) + (Q4% \* Q4 Weight)**
  \* Where Q1%, Q2%, Q3%, and Q4% are the profit percentages for each quarter.
  \* The weight for each quarter is typically 1/4 (since there are four quarters in a year).
**4. Calculation:**
\* Total Profit Percentage = (0.90 \* 0.25) + (0.92 \* 0.25) + (0.81 \* 0.25) + (0.88 \* 0.25)
\* Total Profit Percentage = 0.225 + 0.23 + 0.2025 + 0.22
\* **Total Profit Percentage = 0.8775 or 87.75%.**
**Therefore, the total profit percentage over the year is 87.75%.** |
| **Offline+Humanline** | Here's how to calculate the total profit percentage over the year:
**1. Understand it's not a simple sum:** You can't just add 90% + 92% + 81% + 88% = 351% and divide by 4. Percentages represent portions of the whole, and simply adding them like this would imply profits exceeding 100%, which isn't possible.
**2. Weighted Average Approach:**
\* **Think of each quarter's profit as a weighted contribution:** The weight is the proportion of the year that quarter represents (1/4 or 25%).
\* **Calculate the weighted profit for each quarter:**
  \* Q1: 90% \* 0.25 = 22.5%
  \* Q2: 92% \* 0.25 = 23%
  \* Q3: 81% \* 0.25 = 20.25%
  \* Q4: 88% \* 0.25 = 22%
\* **Add the weighted profits:** 22.5% + 23% + 20.25% + 22% = **87.75%**
**Therefore, the total profit percentage over the year is 87.75%.** |
| **Online** | Here's how to calculate the average profit percentage over the year:
1. **Add up the percentages:** 90% + 92% + 81% + 88% = 351%
2. **Divide the total by the number of quarters:** 351% / 4 = 87.75%
**Therefore, the average profit percentage over the year is 87.75%.**
**Important Note:** This is an **average**. It doesn't mean the company **made** a profit of 87.75% overall. It shows the average **across** the quarters, which were individually quite variable. |

