# OpenReview forum: "Humanline: Online Alignment as Perceptual Loss"
_ICLR.cc/2026/Conference — ICLR 2026 Poster_

### Official Review · Reviewer_xZQd · 2025-10-26

**Soundness:** 3
**Presentation:** 3
**Contribution:** 3
**Rating:** 6
**Confidence:** 2

**Summary:**

The paper studied the problem of reasons and explanations of online alignment better than offline alignment. Through the lens of the prospect theory framework, the authors provided a human-centric explanation. By proving PPO/GRPO in online on-policy clipping recovers a perceptual bias, the authors showed that online alignment acts as a perceptual loss. With these findings, the post-training is more efficient without restriction on the data source and without the cost of performance. Finally, they also test their results in an unverifiable reward setting to follow open-ended instructions and a verifiable reward setting for the math reasoning task.

**Strengths:**

1. The perspective to explain online alignment is interesting by adopting some methods from behavioral economics.
2. The writing of the paper is clean. The authors clearly state the problem, methods, and their results.
3. Most of the related work is cited and discussed. And the authors provided additional related work in Appendix A.
4. The paper provided both theoretical results and empirical results.

**Weaknesses:**

1. The assumption is strong, e.g., Assumption 4.3.
2. Some statement is not rigorous. For example, "If the success of PPO/GRPO can be ascribed to them being perceptual losses".

**Questions:**

1. How is Assumption 4.3 in practice, especially "The cumulative probability of outcomes with higher absolute surprisal than $z_i$ is negligible"? Is this assumption also used in previous work? How did your experimental setting satisfy this assumption?
2. What is the intuition of Definition 4.5 to propose humanline sampling?
3. The term perception plays a central role in your framing, yet its precise mathematical definition is unclear. Could you provide a more formal explanation?

---

> ### Author Response · Authors · 2025-11-25
>
> Thank you for your review! We will do our best to address your questions and concerns:
>
> **Assumption 4.3**: What we list as assumptions are standard for online alignment; they are better thought of as typical conditions and we’ve updated the proof to clarify this. In the context of LLM alignment, Assumption 4.3 can be interpreted as: given the token vocabulary, the vast majority of possible output sequences is nonsensical or irrelevant, which holds trivially. Only a small minority of possible completions are notably good or bad that we want to explicitly align towards or away from them, and the cumulative probability mass of completions that are noteworthy enough to be better/worse than what we’re aligning on is negligible. For example, if we had a random sequence of n tokens, then we wouldn’t treat that as a positive or negative example during alignment — there is no need to explicitly push the policy toward it or away from it; the goal of pretraining and SFT is to avoid producing such sequences to begin with.
>
> **Statement about Perceptual Losses**: This is a conditional statement, not a declarative one. We’re saying if PPO/GRPO are perceptual losses, then we can source training data from anywhere: we go on to show that this is in fact the case for instruction-following and mathematical reasoning.
>
> **Humanline Sampling**: The intuition behind humanline sampling is that we want to use rejection sampling to mimic sampling from the human-perceived distribution. This gives us a rejection criterion where instead of having a uniform random variable, as in traditional rejection sampling, we have a Beta random variable that incorporates the bias from human perception. We also have two criteria because in an online setting, both the reference and policy models change, and the objective probabilities that are fed into a human observer’s capacity function could reflect exposure to either the current policy or the previous one.
>
> **Definition of Perception**: We have defined perception specifically as it relates to the human perception of probability as defined in prospect theory. The formal definition is given in Section 3, specifically in Definitions 3.1 and 3.2. When we say that PPO/GRPO are perceptual losses, for example, we mean that they implicitly incorporate a distortion of probability that is similar to the one humans are said to have in prospect theory.
>
> We hope that we were able to address your concerns! If so, we hope you’ll consider increasing your score. If you have any further questions or need additional information, please let us know, and thank you again for your feedback.

---

### Official Review · Reviewer_r7Xi · 2025-10-31

**Soundness:** 3
**Presentation:** 3
**Contribution:** 3
**Rating:** 6
**Confidence:** 3

**Summary:**

This paper proposes a human-centric explanation of why online, on-policy alignment (e.g., GRPO/PPO) often outperforms offline, off-policy alignment (e.g., DPO/KTO). The key idea is to model human utility via prospect theory and include probability weighting (capacity) in addition to the subjective value. From this, the authors argue that sampling from the current policy better matches the human-perceived outcome distribution, giving intuition on why online > offline.

On the practical side, they introduce a two-part humanline design pattern applicable to DPO/KTO/GRPO: Humanline syncing and Humanline clipping. Empirically, humanline variants trained on offline data close the observed performance gap with online variants on instruction-following and allow 64× less frequent sampling in math-reasoning. Ablations suggest syncing provides the bulk of the gain; upstream clipping adds a smaller but positive effect.

**Strengths:**

**S1. Originality:** Extends alignment as prospect-theoretic optimization by explicitly modeling probability weighting (not just value).

**S2. Practicality:** The humanline recipe (syncing and upstream asymmetric clipping) is simple to implement and integrate into existing algorithms.

**S3. Quality/robustness:** Ablations isolate the roles of syncing vs clipping; results hold across tasks and model sizes.

**S4. Significance:** If adopted, the approach can lower alignment cost and increase offline data usage without sacrificing quality.

**Weaknesses:**

**W1. Dependence on offline data.**
The method depends on the choice of offline dataset (e.g., offline data generated by Gemma-9B vs Llama-8B). The paper attributes this to a possible violation of Assumption 4.1 (support overlap and bounded likelihood ratio), but never verifies it empirically. It would strengthen the work to measure these assumptions directly. For instance, by reporting a coverage or divergence metric and examining how they correlate with performance across datasets.

**W2. Gains stem from syncing, not clipping.**

The ablation study shows that most performance improvements come from syncing the reference model ($\pi_{\text{ref}}$). In contrast, the proposed clipping mechanism, a central contribution of the paper, performs comparably to the offline baseline when used alone. This suggests that clipping contributes little to the observed gains while adding two extra hyperparameters and complexity.

**W3. Unfair comparison to trust-region baselines.**

The comparison to trust-region methods is not hyperparameter-fair. The baseline’s update frequency $k_{tr}=1024$ was taken directly from the original paper, where it was tuned for a different task. In contrast, the proposed method’s equivalent parameter $k$ is shown to be highly sensitive and was tuned for this work. The trust-region baseline could likely perform better if $k_{tr}$ were tuned under the same conditions, so the reported advantage may be overstated.

**W4. Weak theoretical justification.**
Proposition 3.4 only suggests that a small KL divergence is beneficial; it does not theoretically prove that online > offline.

**Questions:**

**Q1.** Could the authors provide quantitative results (e.g., support overlap or KL divergence) to verify when Assumption 4.1 holds and how it relates to performance differences when using different datasets?

**Q2.** See other questions in the weaknesses section.

---

> ### Author Response · Authors · 2025-11-25
>
> Thank you for your review! We appreciate you pointing out the originality, robustness and significance of our work, and we’ll do our best to address your questions and concerns below:
>
> **W1. Offline Data Metric**: We’ve added some additional experiments to the paper in section 5.2.1., finding that the average token log-probability of y|x under the reference model at t=0 is strongly predictive of how useful the offline data will be for alignment, as quantified by downstream performance. For example, if you break up an instruction-following dataset into quartiles based on the average token log-prob at t=0, then training on the lowest quartile of data (in the range [-1.03, -0.36]) leads to performance decline of roughly 20% in relative terms. This isn’t because the lowest quartile of data is off-policy per se, but rather because the less likely it is, the more ratios will be clipped, leading to fewer updates and demanding more examples to reach the same level of performance (all else constant). Because this can be calculated before training even begins, it can be applied either as a filter or as a tool to find higher-quality data.
>
> **W2. Syncing vs. Clipping**: We have clarified the caption for Figure 5. For KTO and GRPO, roughly one-third of the gains of using the humanline variant come from clipping. Without humanline clipping, the gap between offline and online alignment could not be entirely bridged. It is not surprising that clipping without syncing is not that helpful because there is little to be clipped if the reference model does not move. Moreover, the default clipping range ([-1.5, 1.5]) works consistently across many different settings, suggesting that even though there are two new hyperparameters, little hyperparameter tuning is required.
>
> **W3. Comparison to Trust-Region Baseline**: We verified for ourselves — by doing a hyperparameter sweep — that the baseline’s recommended setting for the update frequency was also optimal for our experiments; we therefore do not believe the reported advantage is overstated.
>
> **W4. Proposition 3.4**: This proposition doesn’t say anything about online being better than offline. In plain English, it is simply saying that if we want the absolute difference between our estimate of human utility and the actual utility to be bounded by some threshold, then it suffices for the KL divergence between the distribution used for sampling outputs and the human-perceived distribution to be smaller than some other threshold.
>
> **Q1**. Definitely! As noted above in the reply to W1, the average token log probability under the reference model (at step 0) is a good predictor of whether the offline data, when used with a humanline variant, will be enough to bridge the gap with online alignment (see section 5.2.1).
>
> **Q2**. See our responses above.
>
> We hope that we were able to address your concerns! If so, we hope you’ll consider increasing your score.
>
> If you have any further questions or need additional information, please let us know, and thank you again for your feedback.

---

> > ### Comment · Reviewer_r7Xi · 2025-11-25
> >
> > I thank the authors for the additional experiments and clarifications. W1–W3 are addressed. Regarding W4, I agree the clarification that Proposition 3.4 technically "doesn’t say anything about online being better than offline," the confusion arose from the text following it (Line 241), which states: "This offers a human-centric explanation for why online on-policy sampling is superior to offline off-policy sampling for alignment". I will maintain my acceptance score with increased confidence.

---

### Official Review · Reviewer_fVrX · 2025-10-31

**Soundness:** 3
**Presentation:** 3
**Contribution:** 2
**Rating:** 4
**Confidence:** 3

**Summary:**

In this paper, the authors provide a theoretical study of the reasons why online alignment is better than offline alignment. In particular, this work shows that online on-policy algorithms are good at approximating human perception. Moreover, the offline and online training have been shown they be identical in maximizing human utility. Motivated by these investigations, the authors propose a new design for the training algorithm, which is evaluated on experiments.

**Strengths:**

1. This work studies an important domain - trying to explain why online algorithms are usually better than offline algorithms.
2. The presentation of this paper is easy to follow.
3. The theoretical and empirical investigations are both provided.

**Weaknesses:**

1. From the experimental results, the humanline indeed can help with offline alignments following the lack of human perceptions. However, it is not clear why the online alignments (the authors have shown that those online alignments have demonstrated the capacity for modeling human perceptions) can also be greatly improved with humanline.

2. The modeling of the value function as well as the utility function is hypothetical, which means that if they reflect the true system is not clear; otherwise, the authors need to verify them.

3. Based on the definitions under prospect theory, the authors provide a new analysis for answering the question about the comparison between the offline and online alignments. However, from another perspective, the offline off-policy is indeed a more challenging process compared to online on-policy, which is more computational cost but involves more exploration opportunities. In this case, how to validate the benefits of online alignments is mostly from the explanation under the prospect theory?

4. As the performance difference between offline and online alignments is also witnessed in large model alignments. I am wondering how to verify that the phenomenon observed for small models can also be extended to large models.

**Questions:**

see weakness

---

> ### Author Response · Authors · 2025-11-25
>
> Thank you for your review! We address the points you raised below:
>
> **Online+Humanline**: Regarding your suggestion that “online alignment can also be improved with humanline, and it’s unclear why”, we would like to clarify that as discussed in section 5.2.1., online+humanline is slightly better than online alone, but this is not statistically significant. To the extent that there is a slight — but not significant — improvement, it would ostensibly be from incorporating the perceptual bias more thoroughly, as predicted by our theory.
>
> **Modeling Value Function**: We would also like to clarify that we aren’t modeling the utility function here, only the weighting function. As we discuss in Section 3, shortly after Definition 3.3, it is impossible to ask for people’s perceived probabilities over a vocabulary space of tens of thousands of tokens. The monetary experiments done by Kahneman and Tversky dealt with outcome spaces that were far smaller, small enough where inferring the human-perceived CDFs was actually tractable. This is why we explicitly state this as an assumption, one that we feel is valid because of how well prospect theory has replicated over the recent decades (in contexts with smaller output spaces for the random variables).
>
> **Online Alignment Benefits**: We don’t fully follow the question here, but we believe you are asking “how do we know that this theory exclusively explains why online > offline”? As we note in the paper, there are many theories for why online > offline for alignment. Most of them are RL theoretic; only ours provides a human-centric perspective. We don’t claim anywhere in the paper that this is the only reason why online alignment works so well, but it is a novel explanation that complements – not supplants — all the RL theoretic ones.
>
> **Scaling the Results**: As noted in the paper, due to academic research budgets, we were only able to test our approach up to 27B, which is as far as scaling usually goes in academia. Whether these results will hold at production scale remains to be seen, but we believe that the pattern we’ve seen thus far will continue
>
> We hope that we were able to address your concerns! If so, we hope you’ll consider increasing your score. If you have any further questions or need additional information, please let us know, and thank you again for your feedback.

---

> > ### Comment · Reviewer_fVrX · 2025-11-27
> > **Thank you for the response**
> >
> > I thank the author for their responses, which resolve my previous confusion about the paper. I have no further comments in this stage, and would like to increase the score to 6.

---

### Official Review · Reviewer_Thz1 · 2025-11-02

**Soundness:** 3
**Presentation:** 3
**Contribution:** 3
**Rating:** 4
**Confidence:** 4

**Summary:**

This paper talks about why online on-policy methods tend to outperform offline off-policy methods, and offers a human-centric answer grounded in prospect theory. Specifically, the authors claim that (1) online on-policy sampling better approximates the human-perceived distribution of what the model can produce, and (2) PPO/GRPO recovers a perceptual bias in how humans perceive probability. They then propose a practical humanline design pattern applicable to DPO/KTO/GRPO: (1) humanline syncing (sync the reference with the previous policy every k steps) and (2) humanline clipping (asymmetric clipping of token log-ratio). Empirically, the authors show that offline + humanline matches online performance on instruction following and allows 64-times less frequent sampling without degradation on math reasoning, which gives large speedups versus standard online training.

**Strengths:**

## Originality
1. The paper reframes the online/offline gap through prospect theory from behavioral economics, which is quite a unique point of view. Though it is not the first paper to extend prospect theory to alignment, it considers the probability weighting and the inverted-S curve shown in the paper intuitively explains why online sampling is closer to the human-perceived distribution.
2. The paper theoretically links GRPO/PPO clipping to perceptual weighting via rejection sampling and limit arguments. This gives a fresh interpretation of a widely used heuristic.

## Quality
1. All claims are backed with theoretical proof.
2. Empirically, the paper gives a practical yet minimal recipe to highlight the change. Such a plug-and-run recipe can be useful for future researchers.

## Clarity
The presentation is clear and easy to follow. There is a good example throughout the prospect theory section making it easy to understand.

## Significance
1. If robust, the result meaningfully relaxes the need for fully online training. This would significantly improves the overall training walltime while maintains the same level of performance, greatly speed up the model iterations.

**Weaknesses:**

1. Lack of implications of assumptions. The authors did not cover enough discussion on assumptions 4.1-4.3, e.g., what do those assumptions imply in the language of LLM alignment? Especially for 4.1, same support and finite likelihood ratio seem too restrict in LLM as the action space is enormous; for 4.3, does LLM usually gives you light tail? Any data/study support?
2. Offline data quality dependency. The paper also shows that not all data could match the online performance, but stops discussing more on the implication and in practice how this could be resolve or any efficient ideas to detect an offline dataset with "good quality". Within good/bad data, do they share some common attributes?
3. Data–method confound. Given the offline data quality dependency, can naive offline methods without humanline achieve comparable performance by a carefully selected set of data? If so, then how can we verify the efficacy of humanline? Some carefully designed abalation tests on this would be good.

**Questions:**

See weakness. Happy to raise the score if authors could help clarify on the above issues.

---

> ### Author Response · Authors · 2025-11-25
> **Clarifying Assumptions and Measuring Offline Data Quality**
>
> Thank you for your review! We appreciate you recognizing the originality and significance of our work. We address the points you raised below:
>
> **Implications of Assumptions**: What we list as assumptions are standard for online alignment; they are better thought of as typical conditions, and we’ve updated the proof to clarify this. In the context of LLM alignment, they can be interpreted as follows:
> - 4.1: The action space here is the token vocabulary, so the policy and reference having the same support and finite likelihood ratio just means: that they share a vocabulary (which holds because they are identical at t=0); and that each possible token has non-zero probability, which arises trivially from a softmax output distribution.
> - 4.2: The capacity functions having the same form means that the human-biased CDF has the same structural form for all outcomes. This was found to hold in the original Kahneman and Tverksy’s experiments (for probability in a more general sense).
> - 4.3: This means that the vast majority of possible output sequences is nonsensical or irrelevant, which holds trivially. For any given prompt, the possible output space of n-length sequences is very large. Only a small minority of possible completions are good or bad enough that we want to explicitly align towards or away from them, and the cumulative probability mass of better or worse completions is negligible.
>
> **Offline Data Quality**: We’ve added some additional experiments to the paper in section 5.2.1., finding that the average token log-probability of y|x under the reference model at t=0 is strongly predictive of how useful the offline data will be for alignment, as quantified by downstream performance. For example, if you break up an instruction-following dataset into quartiles based on the average token log-prob of the output at t=0, training on the lowest quartile of data (in the range [-1.03, -0.36]) leads to performance decline of roughly 20% in relative terms. This is less an issue of off-policy-ness and more one of sample efficiency. The less likely the data is, the more the probability ratios will be clipped, leading to smaller updates and requiring more examples to reach the same level of performance (all else constant). Because this metric can be calculated before training even begins, it can be applied either as a filter or as a tool to find higher-quality data.
>
> **Data-Method Confound**: If we take the same filter we used to judge offline data quality in the previous paragraph and do vanilla offline alignment on each quartile, we find that it is only marginally better than when we train on all quartiles (see: Figure 1 vs. Figure 6). That is, even using high-quality offline data is not sufficient to fully bridge the offline-online gap in the absence of humanline methods, though it does help slightly. Naturally, evidence of absence != absence of evidence, but the fact that there exists a simple quality filter for offline data that is predictive of humanline performance but is unable to bridge the online-offline gap on its own suggests that there isn’t a data-method confound at work here.
>
> We hope that we were able to address your concerns! If so, we hope you’ll consider increasing your score. If you have any further questions or need additional information, please let us know, and thank you again for your feedback.

---

> > ### Comment · Reviewer_Thz1 · 2025-11-27
> >
> > Appreciate the authors for the prompt response! The newly added experiments in Section 5.2.1 look great to me, especially the average token logp under the ref model, which provides great insight for how to pick a good dataset for training.
> >
> > However, I still have concerns on Assumption 4.1.
> >
> > > so the policy and reference having the same support and finite likelihood ratio just means: that they share a vocabulary (which holds because they are identical at t=0);
> >
> > This is indeed true.
> >
> > > and that each possible token has non-zero probability, which arises trivially from a softmax output distribution.
> >
> > I would not say this statement is not correct, as the softmax function does guarantee that the token prob is strictly positive. However, it does not guarantee that the likelihood ratio between the trained policy and reference policy is lower/upper bounded by a *constant* (think about an extreme case with a uniform distribution and a delta function, their likelihood ratio can be as large as the size of the vocab) as required in the paper, i.e., $M'$ in Prop 4.1 (or $M_P$, $M_R$ in the proof of Theorem 4.3). I'm not sure if this will affect the results/proofs.
> >
> > Meanwhile, Assumption 4.1 (or condition 1 in page 18 of the revision) is imposed on $\pi_\theta$, which: (1) is a moving target throughout the training process; (2) is a random variable as $\theta$ at least would involve some sampling randomness. This does not seem a clean statement to me, because it implicitly contains a covering number of the space that $\theta$ lives with union bound (if you consider the moving $\theta$ and randomness).  I'm not sure if authors can address this theoretically, i.e., only impose assumptions on deterministic and non-changing variables and then derive the finite likelihood ratio bounds for all possible $\theta$s, but one practical illustration might be to study the sequence of likelihood ratios along the training process, i.e., plot the likelihood ratios of $\pi_\theta$ and $\pi_{\text{ref}}$ as $\theta$ changing during the training, to see those ratios are indeed lower/upper bounded by some reasonable constants, or at least they don't have a trend towards 0 or $\infty$.
> >
> > Given the current status, I will keep my score.

---

> ### Author Response · Authors · 2025-11-27
> **Clarifying the Assumption / Condition**
>
> Thank you for your reply! We appreciate the detailed clarification and now see more clearly where the confusion comes from.
>
> 1) Because the policy and reference models yield softmax distributions over a finite shared vocabulary, for any fixed $(x,y_{<t})$ the likelihood ratio for a given (policy, reference) pair takes finitely many positive values, so there exists a finite bound $M^′_\theta$. Our results only require that this maximum exists; they don't require M′ to be small, known, or independent of vocabulary size. In the proofs, we denote this per-$\theta$ constant as $M'$ for notational convenience. The “uniform vs. delta” example lies outside our regime: a true delta distribution assigns probability 0 to most tokens, so the likelihood ratio is infinite on those tokens and the two distributions do not share support.
>
> 2) Throughout, statements are conditional on a _fixed pair_ of (policy, reference) models. For any such pair realized during training, the softmax parameterization over a finite shared vocabulary ensures that there exists a finite constant $M^′_\theta$ that bounds the likelihood ratio for the tokens in that context. Proposition 4.1 and Theorem 4.3 are applied _separately to each such pair_; they don't require a single bound that holds uniformly over all $\theta$ along the training trajectory. In other words, our results only need a finite bound for each fixed $\theta$, not a uniform covering argument over the entire parameter space.

---

> > ### Comment · Reviewer_Thz1 · 2025-11-27
> >
> > Thanks again for the prompt response! Now those assumptions make more sense to me. I will adjust the score to reflect this. I also suggest the authors to revise the manuscript accordingly to further avoid potential confusions.

---

### Meta-Review · Area_Chair_EqfJ · 2026-01-05

**Summary:**

While the initial reception of the paper was lukewarm, the discussion with the authors was well-received and led to a consensus that the paper is above the acceptance threshold. The paper seems to pursue a somewhat alternative, human-centric explanation for the offline-to-online gap, with common concerns related to new models: clarification of modeling assumptions, presentation issues, comparison with previous baselines. My recommendation to accept the paper reflects the positive author discussion and increased support by the reviewers.

**Reviewer Concerns:**

Reviewer Thz1 had questions about modeling assumptions, and especially seemed to have issues with Assumption 4.1. This was resolved positively over the course of a discussion with the authors. The authors promised to incorporate the clarifications in the final version. The reviewer raised their score to a 6.

Reviewer fVrX had questions about the model and the limited scale of the experiments. The reviewer noted that the authors' responses resolved their confusion and raised their score to a 6.

Reviewer r7Xi had questions about the model and the experimental evaluation, especially regarding fairness to baselines and hyperparameter tuning. The concern was resolved, as confirmed by the reviewer in their last message.

Reviewer xZQd seemed to have an issue with Assumption 4.3. However, the issue was minor, since the reviewer score was above the acceptance threshold. The authors responded with a justification. The authors did not confirm whether the justification was satisfactory to them or not.

**Reviewer Scores:**

This paper saw a significant increase in scores over the course of the discussion with the authors.

Reviewers Thz1 and fVrZ raised their evalution from a 4 to a 6 after a productive discussion.
Reviewer r7Xi maintained the positive score of 6, but noted that the productive discussion with the authors gave them increased confidence in the paper.
Reviewer xZQd started from a positive store of 6 and did not participate in the discussion.

---

### Decision · Program_Chairs · 2026-01-26

Accept (Poster)